# Teaching Human Behavior Improves Content Understanding Abilities Of VLMs

**Somesh Singh**⋆ 🅰 🆅🅱 Ⅲⅅ  **Harini S I**⋆ 🅰 🌐  **Yaman K Singla**⋆ 🅰 🆅🅱 Ⅲⅅ

**Changyou Chen** 🆅🅱  **Rajiv Ratn Shah** Ⅲⅅ  **Veeky Baths** 🌐  **Balaji Krishnamurthy** 🅰

🅰Adobe Media and Data Science Research (MDSR)

ⅢⅅIIITD, 🆅🅱SUNY at Buffalo, 🌐CNRL and APPCAIR at BITS Pilani

✉ behavior-in-the-wild@googlegroups.com

## Abstract

Communication is defined as "*Who* says *what* to *whom* with *what effect*." A message from a communicator generates downstream receiver effects, also known as behavior. Receiver behavior, being a downstream effect of the message, carries rich signals about it. Even after carrying signals about the message, the behavior signal is often ignored while training vision language models. We show that training VLMs on receiver behavior can actually help improve their content-understanding abilities. We demonstrate that training VLMs to predict receiver behaviors, such as likes, comments, and replay graphs, which are available at scale, enhances the VLM's performance across a broad range of downstream content understanding tasks. We show this performance increase over 6 types of behavior, 46 different tasks covering image, video, text and audio over 26 benchmark datasets across both 0-shot and fine-tuning settings, outperforming many supervised baselines on diverse tasks ranging from emotion recognition to captioning by upto 150%. We note that since receiver behavior, such as likes, comments, and replay graphs, is collected by default on the internet and does not need any human annotations to be useful, the performance improvement we get after training on this data is essentially free-lunch. We also release BLIFT, our Behaviour-LLaVA IFT dataset comprising 730k images and videos with their receiver behavior collected from multiple platforms on which we train our models to achieve this. The dataset and code are available at behavior-in-the-wild.github.io/behavior-llava.

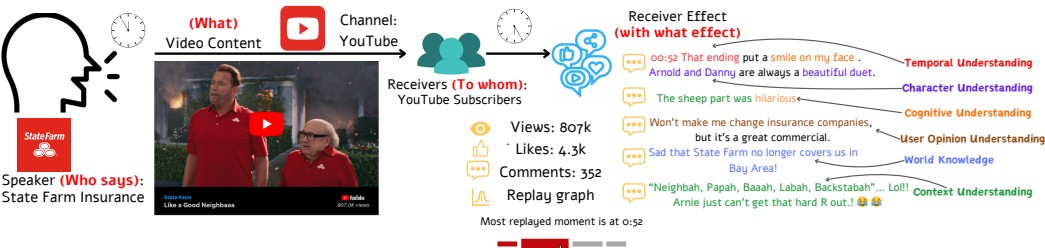

Figure 1: The diagram depicts the five factors of communication in the context of an example YouTube video https://www.youtube.com/watch?v=eT8hO4e2iTM and where lies the free lunch. The receiver effect is not used while training Large Vision and Language Models. However, it contains many important signals that can help in understanding the content. The figure shows several comments containing temporal, cognitive, character, context, and user opinion information useful for understanding the video.

## 1 Introduction

Communication is defined by five factors: sender, message, channel, receiver, and behavior (Shannon & Weaver, 1949; Lasswell, 1948; 1971; Kumar, 2024). Lasswell (1948) encoded these five factors in the phrase, "*Who* says *what* to *whom* with *what effect*." Human behavior occurs as a downstream artifact in the process of communication. Behavior is produced by the receiver as a response to the

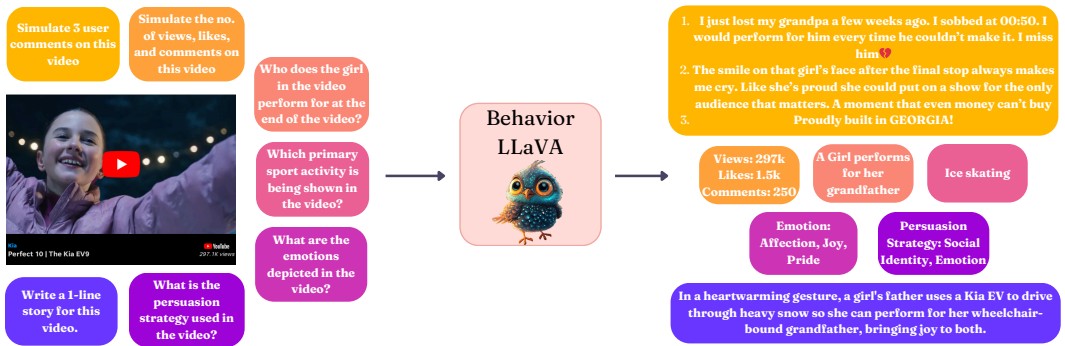

Figure 2: Behavior-LLaVA is trained to answer behavioral questions like simulating user comments and likes on the video. The model, once trained, shows superior performance than LLaMA-Vid and other VLMs on content-related tasks like emotion recognition, action recognition, question answering, persuasion strategy classification, *etc*. The original video was showcased in SuperBowl-2024 and is posted on YouTube on the URL https://www.youtube.com/watch?v=OU7BJc96lI4. The video is titled "Perfect 10: The Kia big game commercial featuring the 2024 Kia EV9" by Kia America.

message sent by the sender. Being a downstream effect, behavior can help us infer important signals about the message itself. These signals, if properly harnessed, should be able to increase performance on the message understanding tasks popular in NLP and CV, like question answering, sentiment analysis, topic classification, *etc*. Despite this, behavior data is considered noise and is ignored while training large language models (Biderman et al., 2022; Penedo et al., 2023) and also large vision and language models (Liu et al., 2023a; Zhu et al., 2023)*. In this paper, we explore this line of thought more.

Humans produce two kinds of behavioral signals upon observing a message (Bertenthal, 1996; Prinz, 1997): perceptual signals and actions as behavior. Perceptual signals, like seeing, touching, and hearing, help a receiver primarily sense the world around her, ultimately guiding her actions. Actions are how a receiver acts on the outside world. The signals produced by the human receiver upon receiving a message carry information about the message itself (Fig. 1). For instance, if a person's heartbeat rises upon watching a movie scene, it can help us infer that perhaps the scene was an exciting scene (Dzedzickis et al., 2020). Similarly, regressing while reading is indicative of important or confusing phrases (Bicknell & Levy, 2011). In these cases, perception behavior helps us derive inferences about content. In a similar vein, the actions a person performs after watching a movie, such as comments and likes, carry signals about the movie (Fig. 1, 2).

Expanding on these ideas, prior literature has shown that harnessing perceptual signals, like eye movements, saliency, keystrokes, mouse movements, and FMRI, by modeling them together with content understanding tasks can improve both NLP and CV tasks. For instance, integration with perception signals causes performance improvement in tasks like visual and natural language question answering (Patro & Namboodiri, 2018; Khurana et al., 2023; Sood et al., 2020), text and image sentiment analysis (Khurana et al., 2023; Barrett et al., 2018; Fan et al., 2018), natural language inference (Khurana et al., 2023), part-of-speech identification (Barrett et al., 2016a;b), named entity recognition (Hollenstein et al., 2019; Hollenstein & Zhang, 2019), syntactic parsing (Plank, 2016), image captioning (Cornia et al., 2018), and visual object detection (Wang et al., 2018b; Kruthiventi et al., 2016).

While the initial studies show that perceptual signals have much promise for improving downstream content understanding, they have a few significant issues due to which integrating human perception has not seen wide adoption in training LLMs. These perceptual signals can only be collected in lab settings requiring specialized lab equipment and are thus expensive to collect and thus are also limited in number. For example, the largest datasets containing the human processing signals are SALICON (Jiang et al., 2015) and Cheng et al. (2014) for visual saliency (10k images each), CELER (Berzak et al., 2022) and Dundee corpus (Kennedy et al., 2013) containing eye movements over 28k sentences and 20 news articles respectively, and Dhakal et al. (2018) containing keystroke patterns over 1.5k

---

*For instance, Pile is a popular dataset used for training LLMs. Biderman et al. (2022) mention that they remove all metadata (containing behavioral data) while creating Pile and subsequently training Pythia. Similarly, behavioral data was removed while creating LLaVA-Instruct-150k (Liu et al., 2023a), MiniGPT (Zhu et al., 2023), and RefinedWeb (Penedo et al., 2023) datasets.

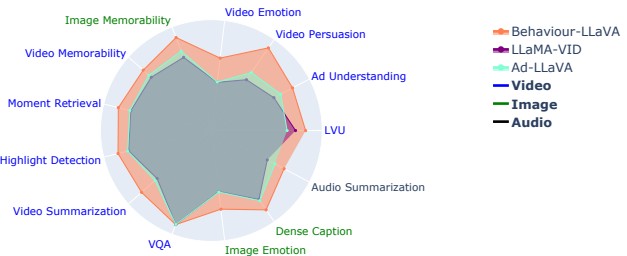

Figure 3: Behaviour-LLava achieves much higher zero-shot performance compared to Ad-LLaVA and the base model LLaMA-VID across a diverse suite of image, video, and audio benchmarks.

sentences. Clearly, these datasets, while making important contributions, do not scale to the level at which today's large language models are trained (trillions of natural language and image tokens).

On the other hand, actions (the other type of behavioral signals produced by a human receiver) are collected at a large scale in the form of digital analytics. Examples of this kind of data are likes, views, shares, comments, and purchase histories on images, tweets, videos, webpages, and other kinds of media. Action data has a much broader representation than is possible in lab settings, is available on more diverse content, and is much cheaper to collect than using specialized lab equipment. At the same time, actions have not been much investigated in the literature for their potential to improve downstream content understanding. Most of the prior literature tries to predict the receiver behavior (Gao et al., 2020; Wang & Torres, 2022; Zhou et al., 2018; Wei et al., 2016; Wang et al., 2018a) rather than using the receiver behavior to improve the content understanding capabilities of models.

Therefore, in this paper, we make initial efforts to collect and understand digital analytics at scale with the aim of integrating them with VLMs to improve their downstream content understanding capabilities. We introduce methods for filtering and cleaning behavioral data and then propose tasks for large language and vision models, leading to improvements in language and visual content understanding tasks. For this, we look to Reddit and YouTube as two major sources of visual content and human behavior in the form of viewer comments, likes, replay graphs, and upvotes. From Reddit, we collect 5 million images and videos along with their upvotes and top-upvoted comments from two major subreddits (r/pics and r/videos). Similarly, from YouTube, we collect 2.2 million videos from 30000 channels along with their likes, views, replay graphs, and top user comments. After extensive filtering and cleaning, we are left with 730k samples of videos and images across the two platforms which we use for the next steps.

After collecting user behavior over image and video content, we design tasks to teach large vision and language models (VLMs) to simulate user behavior. For this, we use an instruction fine-tuning format. Given a video or an image and the other metadata like time of post and channel, we ask the model to simulate user behavior of likes and comments. See Fig 2, Listing 1 for examples. We choose LLaMA-Vid (Li et al., 2023b) as our base model to teach it the user behavior. We call the resultant model Behavior-LLaVA (Large Language and Vision Assistant) (Liu et al., 2023a). We test Behavior-LLaVA on a diverse variety of tasks, evaluating its capabilities on image, video, text, and audio understanding tasks. We compare Behavior-LLaVA against its base model, LLaMA-Vid, and other supervised baselines. Further, to show the impact of behavior, we train another version of LLaMA-Vid, where we train it on the same set of videos and images as Behavior-LLaVA but do not include behavior information. We call this model Ad-LLaVA.

We make the following contributions with this work:

1) **Behavior-LLaVA Instruction Fine-Tuning:** We explore the idea of learning human behavior, resulting in better content understanding. We test this for action-level behavior data such as receiver comments, likes, and replay graphs. We collect a dataset called **BLIFT**, consisting of 400k images and 330k videos, along with their receiver behavior. Then, LLaMA-Vid is trained for the task of predicting receiver comments and upvotes given a media (a video or an image) (Listing 1). We show that using this simple task formulation over behavioral data collected in the wild, results in performance improvement over a hierarchy of tasks. We get improvements over the base LLaMA-Vid across 46 tasks over 26 benchmark datasets in both zero-shot and fine-tuned settings as shown in Figure 3. We show this over low-level content understanding tasks like object and activity recognition and also over high-level tasks like topic and emotion detection. Through this, we propose a scalable

approach to increase the content understanding abilities of VLMs, requiring minimal cost and no architectural changes.

2) **AdLLaVA: Disentangling the effect of content and behavior:** To disentangle the effect of training LLaMA-Vid on additional image and video data from the effect of training on behavior data, we train LLaMA-Vid on BLIFT's videos and images without including behavior. We call this model Ad-LLaVA. We show that Ad-LLaVA shows equivalent performance as its base model LLaVA-Vid; however, Behavior-LLaVA performs better than both Ad-LLaVA and LLaMA-Vid, thus highlighting the importance of behavior data and instruction fine-tuning on behavior data.

3) **Perception vs Action:** We also show an ablation of Behavior-LLaVA across different kinds of behavior. We try out the perception behavior of saliency prediction over images and five types of action-level behavior over images and videos. We find that perception-level behavior does not result in significant performance improvements; however, action-level behavior shows improvements across all the tasks. We posit that one reason for this could be due to the scale for which action-level data is available (Table 10). While perception behavior is mostly collected in lab settings, action-level behavior data is diverse, can be collected in a scalable manner automatically and cheaply.

## 2 METHODOLOGY

In this section, we introduce our approach to train Behavior-LLaVA. Since no publicly available corpus consists of behavior together with image and video content, we first introduce our instruction fine-tuning dataset, "Behavior-LLaVA Instruction Fine-Tuning dataset" (BLIFT). Next, we introduce our methodology to train Behavior-LLaVA. Finally, we report the results of testing Behavior-LLaVA's capabilities on a hierarchy of tasks. The tasks cover low-level media understanding tasks like object and activity detection, high-level media understanding tasks like emotion, topic, and persuasion strategy classification. Figures 14, 15, 17, and 16 illustrate the topic distribution in the BLIFT dataset, highlighting the diversity of image and video themes, as well as comments, identified through BERTopic and GPT-4o-mini clustering.

### 2.1 BLIFT DATASET

Given the abundance of media and behavioral data and its accessibility, our data collection relies on two primary sources: Reddit and YouTube. These platforms share similarities in terms of hosting media content (images and videos) and providing user engagement metrics in the form of Reddit upvotes and comments, and YouTube likes, views, comments, and replay graphs. Here, we delineate the process involved in constructing the instruction fine-tuning dataset, which we term as the Behavior-LLaVA Instruction Fine-Tuning (BLIFT) dataset.

#### 2.1.1 DATA FROM REDDIT

To collect a substantial corpus of diverse images and videos, we targeted at two specific subreddits, namely r/pics and r/videos. Established over 15 years ago, these subreddits had a user base exceeding 20 million during the data collection period, with an average of over 5,000 online users concurrently. Notably, due to stringent content moderation guidelines (Reddit, Inc., 2024; red, 2024a;b;c) and the exclusive focus of these subreddits on media content, they offer a rich variety of content devoid of thematic biases. Our data collection spans until January 2022, during which the activity on these subreddits witnessed a notable decline following several policy adjustments and user protests (Hern, 2023; Economist, 2023).

To ensure data quality and relevance, we executed a series of filtering steps on the posts and comments from these subreddits, illustrated in Figure 13. Initially, we excluded posts predating February 2015 from r/pics, coinciding with the implementation of a rule requiring images without digital/overlay text (red, 2024a;b). This filtering step resulted in the exclusion of 3.1 million images and 2 million videos. Subsequently, considering the sustained popularity of both subreddits, with rankings within the top 20 since 2017 and consistent membership exceeding 20 million, we confined our dataset to posts from January 2018 onwards. This selection process yielded 1.4 million images and 1.1 million videos.

Listing 1: Behavior Instruction fine-tuning template for the video: https://www.youtube.com/watch?v=BKPQkjRF4yY

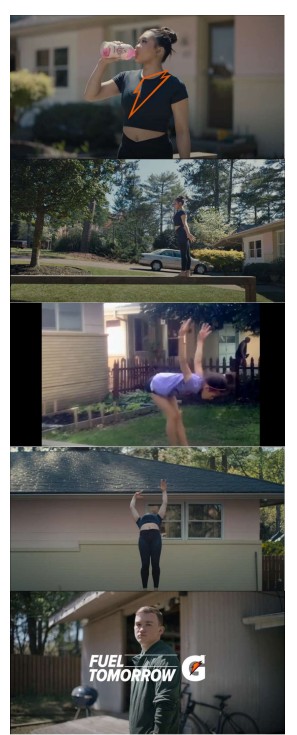

```
<SYSTEM>You are an AI visual assistant. You are given a detailed description of a media,
        followed by the actual media. Answer all questions as if you are seeing the media.</
        SYSTEM>

<USER>
The video advertisement is titled "Gatorade | Make Your Own Footsteps with Suni Lee" for the
        brand Gatorade. The audio in the ad says "[ASR HERE ...]". Analyze this video deeply,
        then write scene by scene description of the video and answer the following questions.
        What percentage of viewers would like this video, and what would be the top-5 popular
        comments on this video? What would the replay graph values for each scene be?
<video>...</video>
</USER>

<ASSISTANT>
The scene-by-scene descriptions are:

Scene 1: The scene shows a woman looking off into the distance with an orange line going
        around the outside of the screen. The foreground colors of the scene are black, mud
        green, gray, dark gray, and the background colors are dark brown, black, dark gray. The
         dominant tone of the scene is neutral. This scene is categorized by the tags: cosmetic
        , eyebrow, face, girl, ponytail, stand, string, woman.
Scene 2: The scene shows a woman balancing on a skateboard in a yard. The foreground colors of
         the scene are black, mud green, dark gray, olive, and the background colors are black,
         dark gray, gray, dark brown.The dominant tone of the scene is neutral. This scene is
        categorized by the tags: athletic, balance, beam, car, girl, house exterior, hurdle,
        jog, legging, plank, rail, seesaw, woman, yard.
Scene 3: The scene shows a girl jumping over a wooden ramp in the backyard. The foreground
        colors of the scene are black, dark gray, gray, dark blue, and the background colors
        are dark brown, dark blue, purple, dark pink, brown.
...

>>> BEHAVIOR <<<

The video will be liked by 2.0% of viewers, and the popular comments could be:
1. "Wow. Love it. She's such an inspiration to the next generation as well as everyone."
2. "Inspiring and great story behind this commercial. Builds meaning and purpose in the hearts
         and minds of youth. It's been a while since good, meaningful ads have been made."
3. "She's an inspiration to the world. Thanks to her,my niece is learning gymnastics.
        Hopefully someday, she is an inspiration to others as Suni is an to everyone"
4. "Chills watching this. Such an inspiration."
5. "Yooooo, this is straight up!"

The replay values for each scene would be:
Scene 1: 0.06
Scene 2: 0.23
Scene 3: 0.38
...
</ASSISTANT>
```

Figure 4: Behavior Instruction fine-tuning template for the video: https://www.youtube.com/watch?v=BKPQkjRF4yY

Further refinement of the dataset involved removing posts and comments marked as NSFW (through community moderators or llama-guard-3-1b (Llama Team, 2024) filters), BOT-generated, or [deleted], along with eliminating duplicate images and videos. This curation step reduced the dataset to 876,000 images and 983,000 videos. To address redundancy in comments, we excluded those comprising fewer than three words and employed TF-IDF-based deduplication with a similarity threshold of 0.6, determined through manual observations.

Following these steps, posts with fewer than two comments were filtered out, resulting in a dataset comprising 631,000 images and 397,000 videos. Additionally, videos exceeding a duration of 500 seconds were omitted, leaving 221,000 videos for analysis. Notably, images not directly hosted on Reddit were excluded due to scraping and copyright limitations. Similarly, for r/videos, only videos hosted on YouTube were considered. It is pertinent to mention that approximately 51% of YouTube videos collected during this period were either made private/unlisted or removed, resulting in 400,000 images and 80,000 videos, accompanied by 1.5 million and 312,000 comments, respectively. These comprehensive filtering steps ensured the construction of a diverse and relevant dataset for fine-tuning instruction-based models.

### 2.1.2 DATA FROM YOUTUBE

Our data collection from YouTube begins with querying Wikidata (Vrandečić & Krötzsch, 2014) for YouTube IDs to compile a list of channels. Wikidata, derived from Wikipedia, provides a curated selection of renowned channels, automatically filtering out noisy videos commonly found in datasets collected from diverse sources like user-generated videos. This initial step yielded a dataset of 2.2

million videos spanning the period from 2018 to 2023, sourced from approximately 6,000 channels collected from Wikidata.

To refine the dataset, manual filtering was employed to exclude certain categories deemed less relevant for our purposes. These categories included music and songs, gaming content, non-English videos, sports commentary, anime, memes, channels with disabled comments sections, and news-related content. Furthermore, and only videos with a substantial viewership, defined as greater than 10,000 views, were retained. We observed that these videos usually have less noisy comments and likes.

Subsequently, the top comments from each video, as ordered by YouTube (i.e., the most liked comments), were selected for inclusion in the dataset. To address redundancy in comments, a TF-IDF filter was applied with a threshold of 0.7, which proved effective in removing duplicate comments prevalent in YouTube data alongside llama-guard-3-1b (Llama Team, 2024) filters.

Comments were further filtered to include only those with a minimum of four words and a maximum of 100 words, ensuring a balance between relevance and conciseness. Additionally, to mitigate the presence of NSFW content, a vocabulary specific to NSFW terms (ldn) was employed to filter out inappropriate posts. On average, we finally get 3.1 comments per video, providing a substantial corpus of user-generated content for analysis. After applying these filtering steps, the dataset was reduced to 250,000 videos, ensuring a curated and relevant collection for subsequent analysis and model training.

## 2.2 Instruction Fine-Tuning LLaMA-Vid

After collecting Reddit and YouTube media and user behavior, we formulate instruction fine-tuning tasks for training LLaMA-Vid. In the training instruction, given the media content and automatic speech recognition if available, we ask the model to simulate the scene-by-scene description and user likes/views and top-5 comments. This instruction training template is given in Listing 1. To generate the instruction data, first, frames are sampled using the 30-degree rule (SI et al., 2023; Arev et al., 2014; Friedman & Feldman, 2004), then the scene-by-scene description are obtained by concatenating automatically generated captions and tags from LLaVA-13B (Liu et al., 2023a), colors and tone through Qin et al. (2020). This instruction format keeps the instructions similar to the instruction format for other VLMs like LLaVA (Liu et al., 2023a), MiniGPT-4 (Zhu et al., 2023), BLIP (Li et al., 2022), LLaMA-Vid (Li et al., 2023b), etc., while additionally teaching the model to learn behavior. We keep the instruction fine-tuning template similar for both YouTube and Reddit. The complete instruction is given in Listing 1.

We start with the trained LLaMA-Vid model. The LLaMA-Vid model uses two tokens to represent each frame in the video, which they call content and context tokens. While the context token encodes the overall image context based on user input, the content token encapsulates visual cues in each frame. For learning context tokens, the model uses attention queries that interact with previously generated image features in the designed attention module. To generate content tokens, the image features are average pooled. This dual-token strategy significantly reduces the number of tokens needed to represent videos, thus enabling the model to scale to longer (hour-long) videos. To better support hour-long videos, LLaMA-Vid was trained on a 9k movie-level conversation instruction set containing plot reasoning and detail understanding questions.

Taking the base LLaMA-Vid model, we finetune it further on behavioral data. Notably, our focus is different from research works such as (Xie et al., 2024; Li et al., 2023c; Chen et al., 2023b) which try to design mechanisms for LLM and VLMs to learn specific behaviors like trust and coordination in game-like synthetic environments. Rather, we teach the VLMs to learn real human behavior and show that it improves downstream performance on content understanding tasks. Therefore, we teach the model that given a media file (image or a video), it has to simulate the likes and comments on the media (Listing 1). In our experiments we observe that the sampling ratio of BLIFT and IFT datasets is an important hyperparameter. We track 4 zero-shot metrics, likes/views, comments perplexity, and empirically find the best results with 1:1 ratio for 2.2 epochs (see Fig 18 and Table 11). For the best checkpoint, the perplexity on comments reduces from 6.22 to 3.05, and the R2 on likes/views goes from -5.1 to 0.45. All experimental hyperparameters are provided in Section E.

We combine 730k instruction pairs from BLIFT with the original instruction tuning dataset consisting of 40K text conversations from ShareGPT, 625K single or multi-turn visual QA pairs, and 98K video

| Model | scene | way_speaking | relationship | like_ratio | view_count | director | genre | writer | year |
|---|---|---|---|---|---|---|---|---|---|
| Video4096-GPT-3.5 generated story + Flan-t5-xxl | 60.2 | 39.07 | 64.1 | 0.061 | 12.84 | 69.9 | 58.1 | 52.4 | 75.6 |
| Video4096-GPT-3.5 generated story + GPT-3.5 classifier | 54.54 | 32.95 | 68.42 | 0.031 | 12.69 | 75.26 | 50.84 | 32.16 | 75.96 |
| LLaMA-Vid + GPT-3.5 Generated Story | 58.12 | 35.5 | 60.6 | 0.314 | 10.34 | 65.34 | 49.77 | 34.23 | 72.12 |
| Ad-LLaVA | 59.05 | 37.07 | 61.2 | 0.319 | 10.37 | 66.84 | 55.13 | 35.33 | 77.34 |
| Behavior LLaVA + GPT-3.5 Generated Story | 66.43 | 41.03 | 64.21 | 0.17 | 5.12 | 71.12 | 63.45 | 39.4 | 79.3 |
| **Improvement of Behavior LLaVA over LLaMA-Vid** | 10.48% | 15.58% | 9.62% | 45.86% | 50.48% | 8.85% | 27.49% | 15.1% | 9.96% |

Table 1: Comparison of various models on the Long Video Understanding benchmark (Wu & Krahenbuhl, 2021) consisting of 9 VQA tasks. We see that Behavior-LLaVA improves on LLaMA-Vid on 9/9 tasks with an average improvement of 21.49%. Further, it outperforms the state-of-the-art in 5/9 tasks.

QA pairs; all the modules except the Visual Encoder are kept frozen. We ablate on multiple sampling ratios from BLIFT. We train the LLaMA-Vid checkpoints with their original SFT mix along with BLIFT. We ablate different sampling ratios and found 1:1 to be empirically performing the best. We train the model for 2.2 epochs, keeping track of the 0-shot evaluation metrics and perplexity on comments in the eval set Fig 18 and Table 11 show the ablations on different sampling ratios and epochs of training. For the best checkpoint, the perplexity on comments reduces from 6.22 to 3.05, and the $R^2$ on likes/views goes from -5.1 to 0.45.

**AdLLaVA to show the impact of behavior data:** To disentangle the effect of training on additional data samples from the effect of training on behavioral data, we train LLaMA-Vid on BLIFT with the video and image verbalization and do not include receiver behavior. Then, the overall instruction template consists of scene-by-scene automatically generated verbalization similar to Listing 1 without the likes and comment simulation. We call the LLaMA-Vid fine-tuned on this data, Ad-LLaVA. We compare Behavior-LLaVA with Ad-LLaVA and LLaMA-Vid along with other state-of-the-art literature benchmarks on various tasks (Tables 1-3).

**Impact of filtering steps on performance:** We use various filtering steps in our data pipeline, including NSFW filters, time filters, bot filters, and gaming and news video filters. To quantify the impact of our filtering steps on the final performance, we compare the performance from training on unfiltered data with filtered data (Figure 18). The figure shows the benefit of our data filtering process. While training on unfiltered BLIFT also improves the result over baseline performance of Llama-Vid but data filtering adds more improvement on top of it

**Ablation with perceptual behavior:** As an ablation experiment, we also try teaching the Behavior-LLaVA perceptual signals. For this, we take the largest perception signal dataset in the literature - Salicon10k (Jiang et al., 2015). It consists of 10,000 MS COCO images (Chen et al., 2015) with free-viewing eye gaze data collected through a novel mouse-based interface. The dataset has been widely used in many studies. We formulate two tasks using this data, (1) **Salicon [Object]**: estimating the saliency over the objects in the image and (2) **Salicon [Region]**: estimating the saliency over a region, where the regions tiles formed by breaking the image into a 3x3 grid. For both tasks we try to model two objectives, ranking and predicting, we found ranking to be much more effective. The instruction are given in Listings 6 and 7.

## 3 RESULTS AND DISCUSSION

In the experimental results, we aim to showcase the diverse and emergent capabilities of our Behavior-LLaVA model through quantitative numbers on various tasks and qualitative examples. These abilities include generating detailed image and video descriptions, emotion and sentiment analysis, question answering, video understanding tasks like scene and action detection. Additionally, we present the ability of Behavior-LLaVA to transfer learn on other behaviors like memorability of a video - both short-term and long-term.

### 3.1 EVALUATION

To test the effectiveness of Behavior-LLaVA, we conduct experiments involving 46 distinct tasks across 26 benchmark datasets. The diversity of tasks and datasets allows us to evaluate the performance and capabilities of Behavior-LLaVA thoroughly. Each of them is covered briefly next:

1. **Visual Question Answering (VQA)**: We evaluate the performance of visual question answering on the following benchmark datasets:
   - The Long-Video Understanding (LVU) benchmark by Wu & Krahenbuhl (2021) comprises nine distinct tasks aimed at assessing long video comprehension, incorporating over 1000 hours of video content. These tasks encompass diverse aspects such as content understanding (including relationship, speaking style, scene/place), prediction of user engagement (YouTube like ratio, YouTube popularity), and movie metadata (director, genre, writer, movie release year).
   - The Holistic Video Understanding (HVU) dataset by Diba et al. (2020) stands as the largest dataset for long video comprehension, comprising 572,000 samples. Encompassing a broad spectrum of semantic elements within videos, HVU tasks involve the classification of scenes, objects, actions, events, attributes, and concepts. Performance evaluation on HVU tasks is conducted using the mean average precision (mAP) metric on the validation set.
   - We also use MSVD-QA, MSRVTT-QA (Chen & Dolan, 2011; Xu et al., 2016b), and ActivityNet-QA (Caba Heilbron et al., 2015) datasets. Their description is given in Appendix D.

2. **Video and Image Understanding Benchmarks**: We use a wide variety of tasks to evaluate video and image understanding: topic, emotion, and persuasion strategy classification, action and reason retrieval and generation, and emotions. We briefly introduce the benchmarks:
   - The advertisements dataset by Hussain et al. (2017) contains 3,477 video advertisements and the corresponding annotations for emotion and topic tags and action-reason statements for each video. There are a total of 38 topics and 30 unique emotion tags per video. Further, we have 5 action-reason statements for each video for the action-reason generation task.
   - Persuasion strategy dataset (Bhattacharyya et al., 2023) is a dataset consisting of 1002 video advertisements from popular brands and their persuasion strategy labels like social identity, anchoring and comparison, reciprocity, foot-in-the-door, *etc.*
   - For emotion analysis, we use VideoEmotion-8 (Asur & Huberman, 2010), Ekman-6 (Xu et al., 2016a), CAER (Lee et al., 2019), IAPSa (Mikels et al., 2005), Emotion6 (Peng et al., 2015), EmoSet (Yang et al., 2023), and Abstract (Machajdik & Hanbury, 2010) datasets. A brief description for each of them is given in Appendix D.

3. **Image Dense Captioning**: Literature image captioning datasets such as MS-COCO (Chen et al., 2015) reduce the inherently rich information and fine-grained semantics to simplistic captions, with very brief statements focussing only on salient objects. Behavior data such as user comments help a model learn much more information such as object and material properties, world knowledge, emotion, character understanding, spatial relationships, aesthetics, *etc.* (see Fig. 1), enhancing the model's captioning capability. Therefore, we design a captioning task to test this capability and compare it with respect to LLaMA-Vid and LLaVA-34B (a 2.5x larger model). Since we do not have ground truth for this task, following the LLM-as-a-judge paradigm, we use GPT-4V as the judge for all the models. GPT-4V is asked to evaluate the dense captions on three metrics: *Correctness* (Listing 2) evaluating the factuality and model hallucinations, *Detail* (Listing 3) evaluating the number and depth of details captured by the generated captions, and *Quality* (Listing 4) measuring the subjective quality of the concepts chosen to be highlighted by the captioning model and the arrangement, coherence, and the linking of various concepts.

4. **Image and Video Memorability Simulation**: Behavior-LLaVA is trained on behavior along with the media. To check if training on behavior helps in solving other behavior tasks (Khandelwal et al., 2024), we test it over image and video memorability simulation. For this, we select seven benchmark datasets covering long-term and short-term memorability over images and videos: LaMem (Khosla et al., 2015), SUN (Isola et al., 2011), and MemCat (Goetschalckx & Wagemans, 2019) for images and Memento10k(Newman et al., 2020), VideoMem (Cohendet et al., 2019), MediaEval (Kiziltepe et al., 2021), and LAMBDA (I et al., 2024) for videos. We briefly cover each of them in Appendix D.

5. **Modalities other than videos and images:** Behavior-LLaVA, built on top of LLaMA-Vid and fine-tuned using BLIFT, is pretrained and fine-tuned on image and video datasets. To test if behavior data can improve the results on other modalities as well, we test Behavior-LLaVA's performance on two tasks across audio and text modalities (Table 12). For audio, we evaluate on the audio summarization task (Han et al., 2023) and for text, we evaluate on the IMDB sentiment benchmark (Maas et al., 2011).

For Tables 1, 5, and 9 we follow the evaluation protocol of Video-4096 (Bhattacharyya et al., 2023), for Table 6 we follow the evaluation protocol of LLaVA and LLaMA-VID. For Tables 2, 3, and 7, for 0-shot evaluation results, we use the logits of the next token from the given task vocabulary. For Table 3, we use the evaluation protocol of SI et al. (2023).

| Training | Dataset | Video Emotion-8 | CAER | Ekman-6 |
|---|---|---|---|---|
| Random | Random | 12.5 | 14.28 | 16.67 |
| 0-Shot | LLaMA-Vid | 29.7 | 27.2 | 37.33 |
| | Behavior-LLaVA | 41.35 | 51.0 | 49.33 |
| | Ad-LLaVA | 29.8 | 27.3 | 37.66 |
| **Improvement of Behavior-LLaVA over LLaMA-Vid** | | 39.22% | 84.19% | 32.14% |
| Finetuned | Zhao et al. (2020) | 54.5 | 78.3 | 55.3 |
| | Zhang et al. (2023c) | 57.3 | 80.1 | 58.2 |
| | eMOTIONS (Wu et al., 2023) | - | - | 53.12 |
| | Arevalo et al. (2017) | 53.7 | 77.3 | 54.2 |
| | Qiu et al. (2020) | 53.3 | - | 57.3 |
| | Xu et al. (2016a) | 52.6 | 77.9 | 55.6 |
| | LLaMA-Vid | 53.8 | 75.6 | 57.9 |
| | Ad-LLaVA | 54.1 | 76.1 | 57.8 |
| | Behavior-LLaVA | 56.9 | 79.3 | 58.4 |
| **Improvement of Behavior-LLaVA over LLaMA-Vid** | | 5.76% | 3.57% | 0.86% |

Table 2: Comparison of various models on three video emotion understanding benchmarks (Video Emotion8 (Jiang et al., 2014), CAER (Lee et al., 2019), Ekman-6 (Xu et al., 2016a)). The main goal of comparing on these benchmarks is to demonstrate Behavior-LLaVA's understanding of complex tasks like video emotions of long-form videos. We see that Behavior-LLaVA improves on LLaMA-Vid on 3/3 benchmarks with an average improvement score of 51.85% in zero-shot and 3.39% in fine-tuned settings. Further, it outperforms the current state-of-the-art on 3/3 benchmarks in zero-shot and 1/3 in fine-tuned settings.

| Training | Models | Image Datasets | | | Video Datasets | | | |
|---|---|---|---|---|---|---|---|---|
| | | Lamem | Memcat | SUN | Memento10k | VideoMem | MediaEval | LAMBDA |
| | Human Consistency | 0.68 | 0.78 | 0.75 | 0.73 | 0.61 | - | 0.61 |
| Finetuned | 10-shot in-context learning GPT-3.5 | 0.29 | 0.18 | 0.15 | 0.07 | 0.06 | 0.06 | 0.06 |
| | ViTMem (Hagen & Espeseth, 2023) | 0.71 | 0.65 | 0.63 | 0.56 | 0.51 | - | 0.08 |
| | Henry trained with 25% data (I et al., 2024) | 0.56 | 0.64 | 0.59 | 0.62 | 0.49 | 0.32 | 0.28 |
| | Henry trained with 50% data (I et al., 2024) | 0.65 | 0.68 | 0.67 | 0.69 | 0.55 | 0.44 | 0.40 |
| | Henry trained with 75% data (I et al., 2024) | 0.71 | 0.75 | 0.73 | 0.74 | 0.62 | 0.49 | 0.47 |
| | Henry trained on all (combined) datasets (I et al., 2024) | 0.72 | 0.79 | 0.76 | 0.72 | 0.60 | 0.48 | 0.52 |
| | Ad-LLaVA trained with 50% data | 0.67 | 0.65 | 0.61 | 0.69 | 0.56 | 0.43 | 0.47 |
| | Behaviour LLaVA trained with 25% data | 0.67 | 0.72 | 0.69 | 0.68 | 0.53 | 0.44 | 0.50 |
| | Behaviour LLaVA trained with 50% data | 0.72 | 0.77 | 0.73 | 0.71 | 0.59 | 0.46 | 0.51 |
| | Behaviour LLaVA trained with 75% data | 0.73 | 0.77 | 0.74 | 0.70 | 0.60 | 0.47 | 0.50 |
| | Behavior-LLaVA trained on all datasets | 0.73 | 0.78 | 0.74 | 0.71 | 0.60 | 0.47 | 0.52 |
| **Improvement of Behavior-LLaVA over LLaMA-Vid (25% data)** | | 19.64% | 12.5% | 16.95% | 9.68% | 8.16% | 37.5% | 78.57% |
| **Improvement of Behavior-LLaVA over LLaMA-Vid (50% data)** | | 10.77% | 13.26% | 8.96% | 2.90% | 7.27% | 4.54% | 27.5% |
| 0-shot | LLaMA-Vid | 0.13 | 0.11 | 0.05 | 0.03 | 0.05 | 0.02 | 0.05 |
| | Ad-LLaVA | 0.14 | 0.13 | 0.06 | 0.06 | 0.07 | 0.04 | 0.13 |
| | Behavior-LLaVA | 0.21 | 0.17 | 0.13 | 0.12 | 0.08 | 0.07 | 0.16 |
| **Improvement of Behavior-LLaVA over LLaMA-Vid** | | 61.5% | 54.5% | 160% | 300% | 160% | 350% | 219% |

Table 3: Comparison of various models on seven video and image memorability benchmarks (Memento10k (Newman et al., 2020), VideoMem (Cohendet et al., 2019), LaMem (Khosla et al., 2015), SUN (Isola et al., 2011), MemCat (Goetschalckx & Wagemans, 2019), MediaEval (Kiziltepe et al., 2021), LAMBDA (I et al., 2024)). The main goal of comparing on these benchmarks is to demonstrate Behavior-LLaVA's understanding of complex and high-level tasks like memorability simulation. We see that Behavior-LLaVA improves on LLaMA-Vid on 7/7 benchmarks with an average improvement score of 186.4% in zero-shot and 39% in fine-tuned settings after seeing 25% train data. Further, it performs similarly to the current state-of-the-art on 7/7 benchmarks in the fine-tuned settings while still seeing only 25% data.

## 3.2 DISCUSSION

Table 13 shows the average 0-Shot performance improvement of Behaviour-LLaVA over Llama-Vid across all tasks. Tables 1, 6, and 9, contain the results for the visual question answering tasks, Tables 2, 5, 7 contain the results for video and image understanding tasks, Tables 8 contains the results for dense-captioning, Table 3 contains the results for image and video memorability benchmarks, and Table 12 contains the results for the audio and text tasks. From the results, we observe the following trends:

- **Behavioral data along with content data causes improvement in content understanding:** A common trend we observe across all the experiments is that Behavior-LLaVA performs better than the base model LLaMA-Vid and the finetuned model Ad-LLaVA on all tasks, especially in zero-shot settings. In fact, Ad-LLaVA performs very similar to LLaMA-Vid itself. This shows that BLIFT adds meaningful signals on average (rather than noise) to the model. Interestingly, the performance gains remain even after fine-tuning on the task dataset (Tables 2, 5, 7).

| Model | Summarization (3 Shot) | | | Moment Retrieval | Highlight Detection |
|---|---|---|---|---|---|
| | BLEU | ROUGE | METEOR | mAP (avg) | mAP |
| Behavior-LLaVA[Replay-Graphs] | 11.2 | 19.1 | 25.3 | **35.2** | **34.9** |
| Behavior-LLaVA[Mem-Recalls] | **11.9** | **20.3** | **26.9** | 34.9 | 33.1 |
| Behavior-LLaVA | 10.6 | 18.3 | 22.7 | 33.1 | 32.7 |
| LLaMA-VID | 9.1 | 16.5 | 20.3 | 30.3 | 32.4 |
| Video-ChatGPT | 5.0 | 14.0 | 19.7 | - | - |

Table 4: Improvement on downstream content understanding tasks by introducing more behaviour signals. Brackets [] denote the new behaviour that we include. Replay graphs (Khandelwal et al., 2024). Mem-Recalls (I et al., 2024) Evaluation done on Multi-shot video summarization (Han et al., 2023) and MomentDETR (Lei et al., 2021)

- **Behavior data leads to more improvements on higher-level tasks:** The performance gains are relatively smaller for low-level tasks of action and object recognition (Tables 6 and 9), but much higher for the more high-level tasks of emotion understanding, sentiment analysis, persuasion strategy classification, and memorability simulation, long form-video understanding and other sub-tasks of Table 9. This indicates that receiver behavior has richer signals for higher-level tasks; in fact, fine-tuned Behaviour-LLaVA models outperform GPT4-V in image emotion recognition. The gains are observed across both image and video benchmarks.
- **Video verbalization leads to more improvement than just using video:** We also observe that classification using a story generated by GPT-3.5 (following Bhattacharyya et al. (2023)) results in better performance than only using the video (Tables 1, 5, 9).
- **Improvement in content understanding capabilities generalize to other modalities:** To evaluate the generalizability of behavior data across modalities, we extend our evaluation to audio summarization and text sentiment analysis and observe improvements of 19.5% (Table 12).
- **Behavior data from BLIFT provides insights into elements like style, topic, emotional content, themes, aesthetics, objects, and other descriptive details (Figs 11,12):** Figures 6, 7, 10, and 5, 8, 9, show several randomly sampled qualitative examples for dense captions generated by Behavior-LLaVA over images and videos respectively. It can be noticed that despite not being explicitly trained for the task of caption generation, the model performs quite well, picking up various artistic, cognitive, and object and material properties. Table 8 contains the results of the quantitative evaluation for the task of dense captioning. The table shows that while Behavior-LLaVA shows a decrease in correctness over LLaMA-Vid, it shows significant improvement in other aspects, including detail and quality. On these aspects, it even comes close to 2.5X larger models (LLaVA-1.6 (34B)).
- **Comparison of improvements observed from perception and action as behavior:** Next, in Table 10, we compare the signals from behavioral data of perception and action. For this, we compare Behavior-LLaVA trained on BLIFT and Behaviour-LLaVA trained on Salicon salient regions and objects. Further, within BLIFT, we compare the performance from predicting singled out behaviours including likes/views, titles, comments. It can be noted that training on just Salicon results in a performance decrease for the lower-level task of action recognition (MSRVTT-QA) but improves on the higher-level task of Emotion recognition. However, the gains are smaller than those observed with training on BLIFT.

## 4 CONCLUSION

In this paper, we explore the idea of learning behavior leading to learning content *better*. Humans produce behavior in response to content. Hence, logically, behavior should contain signals about content, which, if used as a training task, should help in learning content better. We follow this line of thought and show that training large vision and language models on user behavior data of comments and likes collected from Reddit and YouTube leads to performance improvements across a wide variety of tasks. The gains are higher on higher-level tasks such as emotion recognition, persuasion strategy classification, and question answering and smaller on lower-level tasks like action and object recognition. Further, the gains remain even after fine-tuning the VLMs on those benchmarks, thus demonstrating the importance of learning behavior in understanding content better.

## 5 ACKNOWLEDGEMENT

Rajiv Ratn Shah is partly supported by the Infosys Center for AI and the Center of Design and New Media at IIIT Delhi.

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

## APPENDIX

## A    QUALITATIVE EXAMPLES AND DATASET ANALYSIS

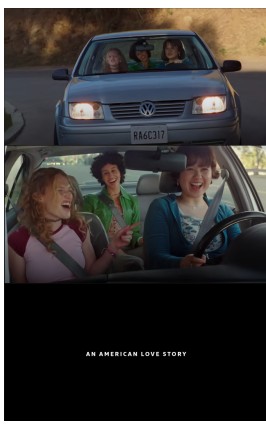

The Volkswagen ad titled begins with a group of three women seated excitedly in a Volkswagen Jetta, the driver sporting a wide smile as she grips the steering wheel. The voiceover sets a nostalgic tone, likening the Jetta to unforgettable first experiences like a first kiss or hearing indie rock for the first time—a symbol of newfound freedom and excitement.

As the scenes unfold, the camera captures the trio cruising down winding roads, their laughter blending with the music and the wind tousling their hair. The atmosphere inside the car is one of camaraderie and adventure, with the Jetta serving as the backdrop to their shared moments of joy and spontaneity. Transitioning to a wider shot of the Jetta gliding along a scenic highway, surrounded by lush greenery, the ad evokes a sense of exploration and the open road. The visuals seamlessly blend modern-day cruising with vintage footage of classic Volkswagen vehicles, reflecting on the brand's 75-year history in America, starting with the beloved Beetle.

The ad concludes with the Volkswagen logo and the tagline "An American Love Story", encapsulating the enduring relationship between Volkswagen and its drivers across generations. This phrase serves as a tribute to Volkswagen's 75-year history in America, beginning with the iconic Type 1 vehicles fondly known as "The Beetle" Through its nostalgic narrative and captivating visuals, the teaser promises viewers an immersive journey into the essence of Volkswagen—a timeless icon that has been a part of countless cherished memories on the American road.

Figure 5: Dense caption generated by Behavior-LLaVA for the video of a Volkswagen ad. The original video is posted at URL: `https://www.youtube.com/watch?v=kyuGXPNr-T0`. The red-colored text highlights the most important aspects of the video captured by Behavior-LLaVA, demonstrating an understanding of aesthetics, characters, world knowledge, emotion, and spatial relationships. More such examples are given in Figs. 6, 7, 10, and Figs. 8, 9 for images and videos respectively.

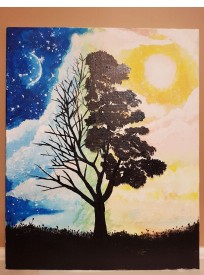

In this captivating painting, a majestic tree takes center stage against a stunning backdrop. The sky above is a poetic split between night and day—a deep, starry blue on one side with a crescent moon, and a warm, radiant blend of yellows and pinks on the other, suggesting either a sunrise or sunset. The tree itself embodies the essence of transition, with one side adorned in lush green foliage and the other stripped bare, symbolizing the timeless dance of seasons and life's perpetual transformations.

This artwork not only captures the eye but also speaks to deeper truths. It invites us to contemplate the delicate balance between light and darkness, growth and dormancy, woven into the fabric of existence. The artist's brushstrokes, rendered with a serene palette and gentle abstraction, evoke a sense of tranquility and introspection. This painting beautifully reminds us to embrace the fleeting beauty of each passing moment, whether bathed in daylight or cloaked in the quiet mystery of night.

Figure 6: Dense caption generated by Behavior-LLaVA for an artistic drawing of a tree.

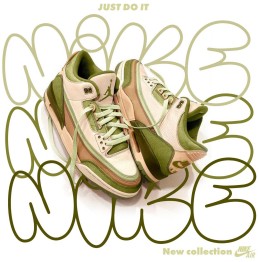

The image features a pair of Nike Air Force 1 sneakers, which are a classic and iconic design. The sneakers are positioned in the center of the image, with the left shoe slightly in front of the right one, giving a sense of depth. They are predominantly white with green accents, including the Nike swoosh logo on the side and the outsole. The laces are white, and the midsole appears to be a light beige color. The background is a plain, light color, which contrasts with the sneakers and makes them stand out. Overlaid on the image is a stylized text that reads "JUST DO IT" in a cursive, flowing script. The text is in a light green color that matches the accents on the sneakers, creating a cohesive color scheme. Below the main text, there is a smaller line of text that says "New collection," indicating that these sneakers are part of a new line or series from Nike.

The overall style of the image is clean and modern, with a focus on the product. The use of the "JUST DO IT" slogan, which is a well-known tagline for Nike, adds a layer of branding and recognition to the image. The text is designed to be eye-catching and to draw attention to the sneakers, which are the main subject of the image. The composition is balanced, with the sneakers centrally placed and the text evenly distributed around them.

Figure 7: Dense caption generated by Behavior-LLaVA for a Nike ad. The red-colored text highlights the most important aspects of the video captured by Behavior-LLaVA, demonstrating an understanding of aesthetics, characters, world knowledge, emotion, and spatial relationships.

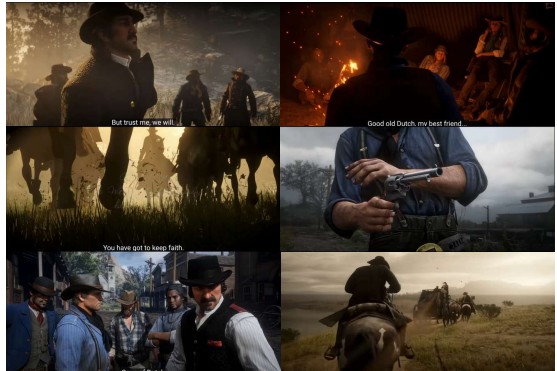

In a heart-pounding and visually stunning trailer for Red Dead Redemption, we are thrust into the gritty world of the American frontier. The trailer opens with a voiceover, a chilling warning delivered with calm certainty: "Listen to me, we don't want to kill any of you... But trust me, we will." Scenes flash by in quick succession, each more intense than the last. We see a lone figure, silhouetted against a setting sun, riding a magnificent horse through a sprawling, golden field. The rugged beauty of the landscape contrasts sharply with the impending sense of danger.

Cut to a dimly lit saloon where a group of hardened men sit around a table, cards in hand, tension thick in the air. The voiceover continues, "This whole thing is pretty much done. We're more ghosts than people."

A flurry of action unfolds: a quick draw in a darkened room, bullets slicing through the air with deadly precision. The voiceover reminisces, "Good old Dutch, my best friend... You know how we met? A pair of hucksters trying to rob each other... Back in '78 or thereabouts."

The visuals intensify as we witness a robbery in progress, chaos erupting as masked figures burst into a bank. "Ladies and gentlemen, this is a robbery," declares one of the outlaws, setting the stage for a clash between lawlessness and order.

Amidst the chaos, snippets of camaraderie emerge: "Sons of Dutch. Makes us brothers." But looming shadows of betrayal and regret cast doubt on these bonds. "Sometimes, brothers make mistakes," acknowledges the voiceover, acknowledging the complexities of loyalty and survival in this unforgiving world.

The trailer crescendos with a crescendo of gunfights, horseback chases, and impassioned speeches. "You'll never change. I know that," declares a voice, capturing the immutable nature of the human spirit amidst adversity. Throughout, a thematic motif resonates: "You have to love yourself a fire." The elemental symbolism underscores the primal nature of existence in a land where survival demands courage and cunning.

As the trailer draws to a close, we're left with the haunting refrain: "Stay strong. Stay with me." The screen fades to black, leaving us hungry for more of this epic tale set against the backdrop of the untamed West.

With breathtaking visuals and evocative storytelling, this Red Dead Redemption trailer promises an unforgettable journey through a world where danger lurks around every corner, and the line between hero and outlaw blurs in the dust and shadows of the frontier.

Figure 8: Dense caption generated by Behavior-LLaVA for the video of the official trailer of the game Red Dead Redemption 2. The original video is posted at URL: https://www.youtube.com/watch?v=eaW0tYpxyp0.

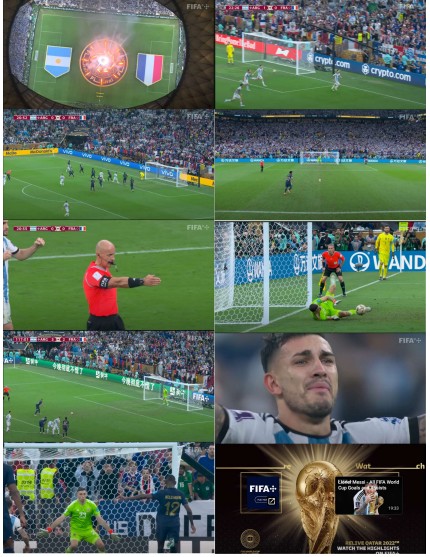

The scene is set at a packed stadium buzzing with excitement, the air thick with anticipation as Argentina and France face off in what promises to be an epic World Cup final. Lionel Messi steps up confidently to take a crucial penalty early in the match, the tension palpable as he eyes the goal. The commentator's voice echoes through the stadium, "He's got the ground, he's got a penalty! A heart beats... And Messi!" The crowd holds its breath as Messi strikes the ball cleanly, sending it into the back of the net. The stadium erupts with cheers as Argentina takes the lead, setting the stage for a thrilling encounter.

As the game unfolds, both teams showcase their attacking prowess. Argentina's Angel Di Maria emerges as a standout, capitalizing on a swift counterattack to score a spectacular goal. "Alvarez, here's McAllister, Di Maria is the spare man, and how Di Maria! Glorious goal! Argentine gold!" The Argentine fans go wild, waving their flags and chanting passionately. However, France fights back fiercely, earning a penalty of their own which they duly convert.

The drama intensifies in the second half as France's Kylian Mbappé dazzles with his speed and skill, scoring a breathtaking equalizer. "Mbappé, to Lamp, Mbappé! Oh wow!" The match swings back and forth, with both teams pushing for victory. In the dying moments of extra time, the score is deadlocked at 3-3 when Argentina's goalkeeper makes a crucial save, denying France a late winner. The match heads to a nail-biting penalty shootout. With nerves of steel, Argentina's players step up and convert their penalties flawlessly, culminating in a decisive save by the goalkeeper. "Argentina, champions of the world!" The stadium erupts once more, as Messi lifts the World Cup trophy amidst a sea of jubilant fans, marking a historic and unforgettable victory for Argentina

Figure 9: Dense caption generated by Behavior-LLaVA for the video of Argentina vs France FIFA World Cup Qatar 2022 Highlights. The original video is posted at URL: `https://www.youtube.com/watch?v=zhEWqfP6V_w`.

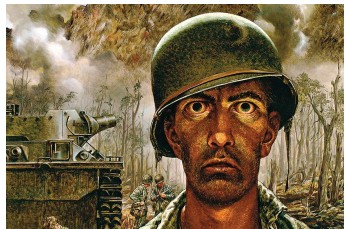

The composition of the image presents a compelling narrative of a soldier immersed in a war-torn landscape. Positioned amidst a backdrop of explosive chaos and dense foliage, the soldier, adorned in traditional military gear, gazes directly at the viewer with a resolute demeanor. The soldier's face is camouflaged, blending seamlessly with the olive-green helmet—a hallmark of battlefield attire.

Surrounding this central figure, a tableau of activity unfolds: a tank looms in the background, its barrel skyward, suggesting recent action. Other soldiers, alert and vigilant, navigate the dense jungle terrain, underscoring the high stakes of the conflict. The foliage, lush yet foreboding, heightens the palpable tension of the scene.

Foregrounded by the silhouette of another soldier's helmeted head, the viewer is drawn into the heart of the action, evoking a sense of shared experience amid the perils of warfare. The realism of the depiction accentuates the emotional weight of the moment, capturing the essence of human resilience amidst the ravages of battle.

Overall, the image transcends mere representation, offering a poignant reflection on the individual's role in the broader narrative of war—imbued with suspense, authenticity, and a profound exploration of the human condition within conflict.

Figure 10: Dense caption generated by Behavior-LLaVA for a painting of a soldier. The model captures many qualitative aspects that are usually missed in common captioning tasks.

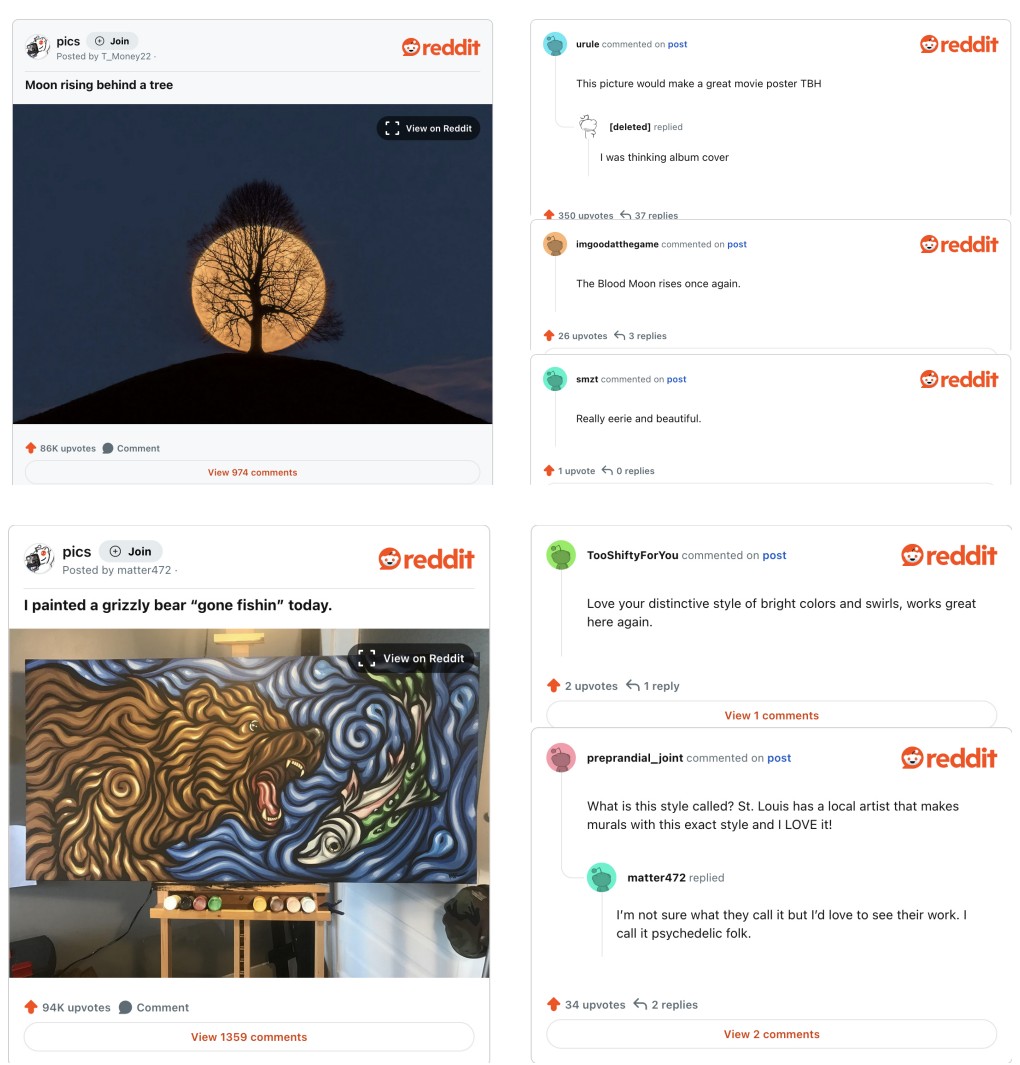

Figure 11: Qualitative examples from r/pics in the BLIFT dataset, accompanied by their corresponding comments. The behavior from Reddit comments provides insights into various elements, including style, topic, emotional content, conceptual themes, object identification, aesthetic qualities, and descriptive details of the post.

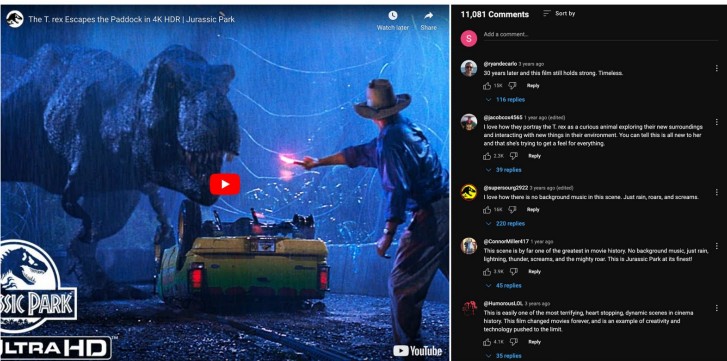

Figure 12: Qualitative example from YouTube along with its comments. The behavior from YouTube comments provides insights into various elements, including style, topic, emotional content, conceptual themes, object identification, aesthetic qualities, and descriptive details.

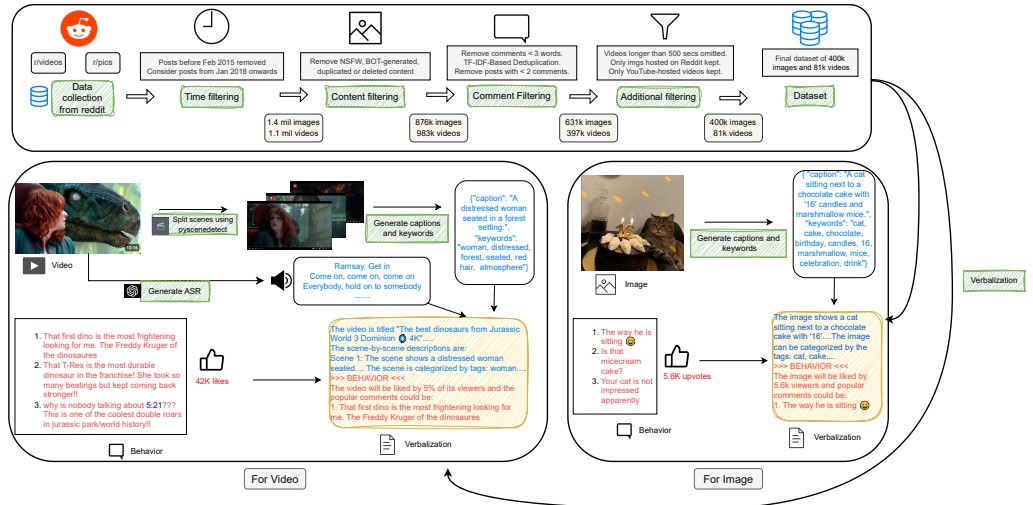

Figure 13: A diagrammatic representation of the process followed to process reddit data. Data was collected from r/pics and r/videos and filtered to ensure quality, excluding older, NSFW, duplicate, and irrelevant content. The final dataset comprises 400,000 images, 80,000 videos, and 1.8M+ comments, focusing on Reddit-hosted and YouTube videos. After filtering the data, we process the collected images and videos by generating captions and keywords for the images. For videos, we segment them into scenes, generate captions and keywords for each scene, and produce ASR transcripts for the entire video. Finally, we combine this information with post comments to create the final prompts.

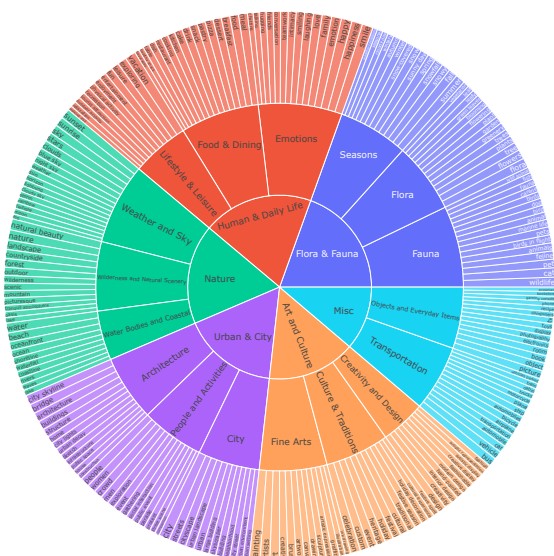

Figure 14: To understand the topic distribution of the images in the BLIFT dataset, we extracted the topics from image titles and captions using BERTopic. Further these topics were clustered and assigned a name by GPT-4o-mini. This figure shows that the BLIFT dataset covers a large and diverse types of image topics and themes.

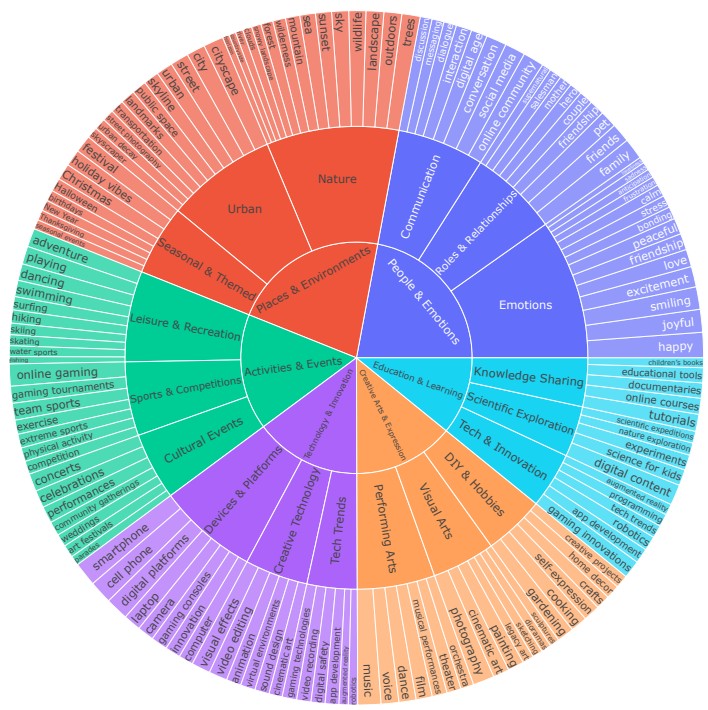

Figure 15: To analyze the topic distribution of receiver feedback we extract topics from the comments of all images using BERTopic. Further these topics were clustered and assigned a name by GPT-4o-mini. This figure shows that the BLIFT dataset covers complex and diverse types of responses including emotions, art styles, themes, concepts and aesthetics.

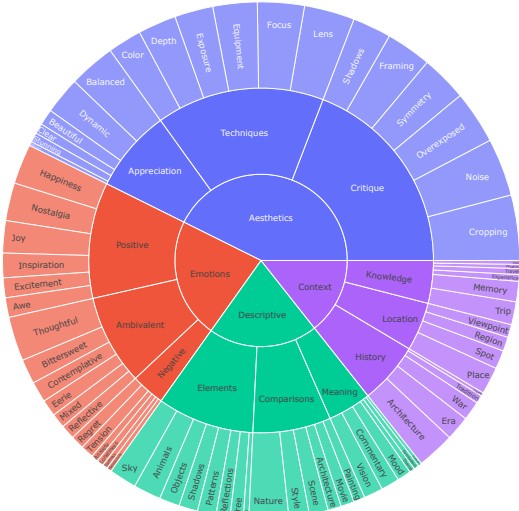

Figure 16: To understand the topic distribution of the videos in the BLIFT dataset, we extracted the topics from video titles, description, scene captions and keywords using BERTopic. Further these topics were clustered and assigned a name by GPT-4o-mini. This figure shows that the BLIFT dataset covers a large and diverse types of video topics and themes.

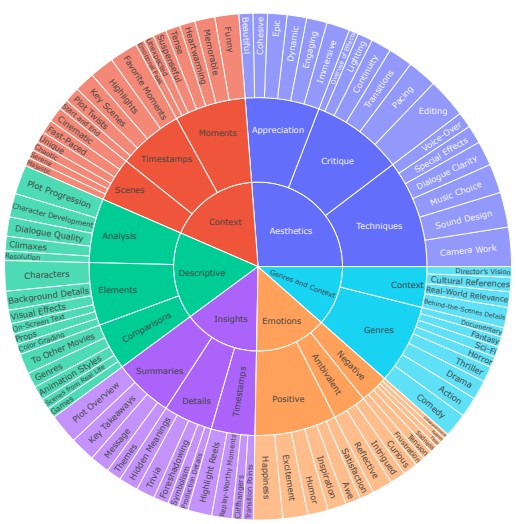

Figure 17: To analyze the topic distribution of receiver feedback we extract topics from the comments of all Youtube and Reddit videos using BERTopic. Further these topics were clustered and assigned a name by GPT-4o-mini. This figure shows that the BLIFT dataset covers complex and diverse types of responses including emotions, art styles, themes, concepts and aesthetics.

# B QUANTITATIVE RESULTS

| Training | Model | Topic | Sentiment | | Persuasion | Action | Reason |
|---|---|---|---|---|---|---|---|
| | | | Clubbed | All labels | | | |
| Random | Random | 2.63 | 3.37 | 14.3 | 8.37 | 3.34 | 3.33 |
| Zero-shot | VideoChat (Li et al., 2023a) | 9.07 | 3.09 | 5.1 | 10.28 | - | - |
| | Video4096 - GPT-3.5 Generated Story + GPT-3.5 Classifier (Bhattacharyya et al., 2023) | 51.6 | 11.68 | 79.69 | 35.02 | 66.27 | 59.59 |
| | Video4096 - GPT-3.5 Generated Story + Flan-t5-xxl Classifier (Bhattacharyya et al., 2023) | 60.5 | 10.8 | 79.10 | 33.41 | 79.22 | 81.72 |
| | Video4096 - GPT-3.5 Generated Story + Vicuna Classifier (Bhattacharyya et al., 2023) | 22.92 | 10.8 | 67.35 | 29.6 | 21.39 | 20.89 |
| | Video4096 - Vicuna Generated Story + GPT-3.5 Classifier (Bhattacharyya et al., 2023) | 46.7 | 5.9 | 80.33 | 27.54 | 61.88 | 55.44 |
| | Video4096 - Vicuna Generated Story + Flan-t5-xxl Classifier (Bhattacharyya et al., 2023) | 57.38 | 9.8 | 76.60 | 30.11 | 77.38 | 80.66 |
| | Video4096 - Vicuna Generated Story + Vicuna Classifier (Bhattacharyya et al., 2023) | 11.75 | 10.5 | 68.13 | 26.59 | 20.72 | 21.00 |
| | LCBM (Khandelwal et al., 2024) | 42.17 | 7.08 | 58.83 | 32.83 | 39.55 | 27.91 |
| | LLaMA-VID w/ only video | 10.11 | 3.42 | 5.75 | 12.32 | 29.61 | 24.11 |
| | LLaMA-VID w/ video + GPT-3.5 Story | 42.72 | 11.05 | 64.02 | 32.07 | 37.76 | 42.33 |
| | Behavior-LLaVA w/ only video | 22.65 | 11.13 | 60.04 | 13.39 | 42.66 | 33.33 |
| | Behavior-LLaVA w/ video + verbalization | 46.34 | 11.7 | 64.13 | 33.33 | 52.06 | 52.03 |
| | Ad-LLaVA w/ video + GPT-3.5 story | 51.16 | 11.33 | 68.03 | 33.11 | 43.26 | 51.45 |
| | Behavior-LLaVA w/ video + GPT-3.5 story | 60.09 | 12.84 | 79.94 | 36.12 | 67.10 | 79.18 |
| | **Improvement of Behavior-LLaVA over LLaMA-Vid** | 40.66% | 16.2% | 24.86% | 12.62% | 77.7% | 87.05% |
| Finetuned | VideoMAE (Tong et al., 2022) | 24.72 | 29.72 | 85.55 | 11.17 | - | - |
| | Hussain et al. (2017) | 35.1 | 32.8 | - | - | 48.45 | - |
| | Intern-Video (Wang et al., 2022) | 57.47 | 36.08 | 86.59 | 5.47 | 6.8 | - |
| | Video4096- Generated Story + Roberta Classifier (Bhattacharyya et al., 2023) | 71.3 | 33.02 | 84.20 | 64.67 | 42.96 | 39.09 |
| | LLaMA-VID w/ video + verbalization | 59.13 | 32.11 | 79.15 | 50.93 | 50.32 | 30.13 |
| | LLaMA-VID w/ video + GPT-3.5 Story | 63.11 | 35.01 | 84.15 | 55.01 | 57.11 | 45.73 |
| | Behavior-LLaVA w/ only video | 58.03 | 22.72 | 84.41 | 26.23 | 59.33 | 51.45 |
| | Behavior-LLaVA w/ video + verbalization | 68.32 | 33.92 | 85.93 | 64.72 | 70.89 | 75.34 |
| | Ad-LLaVA w/ video + GPT-3.5 story | 66.34 | 36.24 | 84.09 | 58.31 | 68.15 | 78.15 |
| | Behavior-LLaVA w/ video + GPT-3.5 story | 71.2 | 39.55 | 86.17 | 65.03 | 80.44 | 81.67 |
| | **Improvement of Behavior-LLaVA over LLaMA-Vid** | 12.82% | 12.97% | 2.4% | 18.21% | 40.85% | 78.59% |

Table 5: Comparison of various models on two video understanding benchmarks (Hussain et al., 2017; Kumar et al., 2023) consisting of 5 tasks related to video advertisements understanding. The main goal of comparing on these benchmarks is to demonstrate Behavior-LLaVA's understanding of complex videos. We see that Behavior-LLaVA improves on LLaMA-Vid on 5/5 tasks with an average improvement score of 43.18% in zero-shot and 27.64% in fine-tuned settings. Further, it outperforms the current state-of-the-art on 3/5 tasks in zero-shot and 5/5 in fine-tuned settings.

| Method | LLM | MSVD-QA | | MSRVTT-QA | | ActivityNet-QA | |
|---|---|---|---|---|---|---|---|
| | | Acc | Score | Acc | Score | Acc | Score |
| FrozenBiLM (Yang et al., 2022) | DeBERTa-V2 | 32.2 | – | 16.8 | – | 24.7 | – |
| VideoLLaMA (Zhang et al., 2023a) | Vicuna-7B | 51.6 | 2.5 | 29.6 | 1.8 | 12.4 | 1.1 |
| LLaMA-Adapter (Zhang et al., 2023b) | LLaMA-7B | 54.9 | 3.1 | 43.8 | 2.7 | 34.2 | 2.7 |
| VideoChat (Li et al., 2023a) | Vicuna-7B | 56.3 | 2.8 | 45.0 | 2.5 | 26.5 | 2.2 |
| Video-ChatGPT (Maaz et al., 2023) | Vicuna-7B | 64.9 | 3.3 | 49.3 | 2.8 | 35.2 | 2.7 |
| BT-Adapter (Liu et al., 2023b) | Vicuna-7B | 67.5 | **3.7** | 57.0 | 3.2 | 45.7 | 3.2 |
| LLaMA-VID | Vicuna-7B | 69.7 | **3.7** | 57.7 | 3.2 | 47.4 | 3.3 |
| LLaMA-VID | Vicuna-13B | 70.0 | 3.7 | 58.9 | 3.3 | 47.5 | 3.3 |
| Ad-LLaVA | Vicuna-13B | 70.0 | 3.7 | 59.0 | 3.3 | 47.4 | 3.3 |
| Behaviour-LLaVA | Vicuna-13B | 70.1 | 3.7 | 59.2 | 3.4 | 47.5 | 3.3 |
| **Improvement of Behavior-LLaVA over LLaMA-Vid** | | 0.14% | 0% | 0.5% | 3% | 0% | 0% |

Table 6: Comparison of various models on three conventional video question answering benchmarks consisting of question answers related to action understanding. The main goal of comparing on this benchmark is to show that Behavior-LLaVA does not perform worse on low-level understanding tasks like action recognition. We see that Behavior-LLaVA marginally improves on LLaMA-Vid on 2/3 benchmarks. Further, it performs equivalent to the state-of-the-art in 3/3 benchmarks.

| Training | Models | IAPSa-8 | Abstract | Emotion6 | Emoset |
|---|---|---|---|---|---|
| Random | Random | 12.5 | 12.5 | 16.67 | 12.5 |
| 0-shot | GPT4-V | 83.33 | 71.12 | 65.47 | 79.16 |
| | LLaMA-Vid | 43.41 | 43.24 | 40.37 | 45.23 |
| | Ad-LLaVA | 43.22 | 43.01 | 43.21 | 44.38 |
| | Behavior-LLaVA | 57.97 | 64.21 | 49.71 | 50.38 |
| **Improvement of Behavior-LLaVA over LLaMA-Vid** | | 33.54% | 48.5% | 23.14% | 11.39% |
| Finetuned | MIDAN (Xu et al., 2022) | 85.96 | 78.34 | 61.66 | 75.75 |
| | Stimuli-aware (Yang et al., 2021) | - | - | 61.62 | 78.40 |
| | LLaMA-VID finetuned | 84.93 | 71.23 | 62.87 | 80.31 |
| | Ad-LLaVA finetuned | 85.13 | 71.16 | 62.66 | 79.88 |
| | Behavior-LLaVA finetuned | 87.36 | 81.41 | 72.31 | 83.21 |
| **Improvement of Behavior-LLaVA over LLaMA-Vid** | | 2.86% | 14.29% | 15.02% | 3.61% |

Table 7: Comparison of various models on four image emotion understanding benchmarks (IAPSa-8 (Mikels et al., 2005) Abstract (Machajdik & Hanbury, 2010), Emotion6 (Peng et al., 2015), Emoset (Yang et al., 2023)). The main goal of comparing on these benchmarks is to demonstrate Behavior-LLaVA's understanding of complex tasks like image emotions. We see that Behavior-LLaVA improves on LLaMA-Vid on 4/4 benchmarks with an average improvement score of 29.14% in zero-shot and 8.95% in fine-tuned settings. Further, it outperforms the current state-of-the-art on 4/4 benchmarks in the fine-tuned settings.

| Model | Correctness | Detail | Quality | Average |
|---|---|---|---|---|
| GPT4-V | 8.4 | 8.5 | 8.4 | 8.43 |
| LLaVA-1.6 (34B) | 8.1 | 8.2 | 7.4 | 7.9 |
| LLaMA-Vid (13B) | 7.4 | 7.6 | 7.2 | 7.4 |
| Ad-LLaVA (13B) | 7.5 | 7.8 | 7.3 | 7.53 |
| Behavior-LLaVA (13B) | 7.3 | 8.1 | 7.9 | 7.76 |
| **Improvement of Behavior-LLaVA over LLaMA-Vid** | -1.3% | 6.57% | 9.72% | 4.8% |

Table 8: Comparison of various models on the image dense captioning task. The main goal of this task is to demonstrate Behavior-LLaVA's image captioning ability. Despite not being explicitly trained on this task, Behavior-LLaVA performs better than both Ad-LLaVA and LLaMA-Vid on Detail and Quality aspects while losing marginally on correctness. On the aspects of detail and quality, it even outperforms the much larger model of LLaVA-1.6 (34B).

| | Model | Scene | Object | Action | Event | Attribute | Concept | Overall |
|---|---|---|---|---|---|---|---|---|
| 0-shot | LLaMA-VID | 47.25 | 65.26 | 78.12 | 28.03 | 42.33 | 50.03 | 51.83 |
| | Ad-LLaVA | 49.10 | 65.35 | 77.45 | 31.45 | 43.33 | 50.70 | 52.91 |
| | Behavior-LLaVA | 52.03 | 65.33 | 77.95 | 32.66 | 45.67 | 51.20 | 54.14 |
| **Improvement of Behavior-LLaVA over LLaMA-Vid** | | 10.12% | 0.11% | -0.22% | 16.52% | 7.89% | 2.34% | 4.46% |
| 0-shot w/ story | Video4096 - GPT-3.5 generated story + Flan-t5-xxl classifier | 59.66 | 98.89 | 98.96 | 38.42 | 67.76 | 86.99 | 75.12 |
| | Video4096 - GPT-3.5 generated story + GPT-3.5 classifier | 60.2 | 99.16 | 98.72 | 40.79 | 67.17 | 88.6 | 75.77 |
| | LLaMA-VID + Generated Story | 60.3 | 99.92 | 99.01 | 39.33 | 66.66 | 87.33 | 75.425 |
| | Behavior-LLaVA + GPT3.5 generated story | 60.4 | 99.89 | 98.23 | 40.97 | 67.23 | 88.33 | 75.84 |
| **Improvement of Behavior-LLaVA over LLaMA-Vid** | | 0.16% | -0.03% | -0.78% | 4.17% | 0.85% | 1.14% | 0.55% |

Table 9: Comparison of various models on the Holistic Video Understanding benchmark (Diba et al., 2020) consisting of 7 VQA tasks. We see that Behavior-LLaVA improves on LLaMA-Vid on 6/7 tasks with an average improvement of 5.88%. Further, it performs equivalent to the state-of-the-art in 6/7 tasks. State of the art is achieved by generating a story and asking Behavior-LLaVA to answer questions based on the generated story.

| Task | MSRVTT-QA | CAER | Emoset | Comments | likes/views |
|---|---|---|---|---|---|
| Base | 58.9 | 75.6 | 45.23 | 6.22 | -0.1 |
| Salicon [Region] | 55.6 | 75.8 | 47.25 | 6.25 | -0.07 |
| Salicon [Object] | 57.9 | 76.4 | 48.0 | 6.12 | 0.05 |
| BLIFT[Likes/Views] | 58.2 | 76.2 | 47.12 | 6.15 | 0.38 |
| BLIFT[Titles] | 58.4 | 78.1 | 48.12 | 5.09 | 0.13 |
| BLIFT[Comments] | 59.0 | 79.1 | 49.58 | 3.02 | 0.19 |
| BLIFT | 59.2 | 79.3 | 50.38 | 3.05 | 0.40 |

Table 10: Ablation on using comments and/or perception signals from Salicon

| Sampling Ratio | Epoch | Likes/Views $R^2$ | Comments Perplexity | Performance |
|---|---|---|---|---|
| Base-Model | 0 | -0.1 | 6.22 | 0 |
| 1:1 | 0.5 | 0.11 | 4.71 | 5.49 |
|  | 1 | 0.22 | 3.95 | 8.23 |
|  | 1.25 | 0.33 | 3.19 | 10.97 |
|  | 1.5 | 0.35 | 3.13 | 11.79 |
|  | 2 | 0.38 | 3.08 | 12.31 |
|  | 2.2 | 0.4 | 3.05 | 12.57 |
| 1:2 | 0.5 | 0.14 | 4.33 | 3.04 |
|  | 1.05 | 0.28 | 3.66 | 7.08 |
|  | 1.45 | 0.42 | 2.99 | 8.12 |
| 1:10 | 0.5 | 0.15 | 3.43 | 1.38 |
|  | 0.8 | 0.31 | 2.78 | 3.44 |
|  | 1 | 0.38 | 2.46 | 4.48 |
|  | 1.2 | 0.46 | 2.13 | 5.51 |
| 2:1 | 0.5 | 0.1 | 5.3 | 3.52 |
|  | 1 | 0.21 | 4.6 | 7.45 |
|  | 1.5 | 0.31 | 3.9 | 9.13 |

Table 11: Ablation on different sampling ratios and epochs of training. Sampling ratio is the ratio of behaviour data to multimodal instruct data. Performance is the average increase in 0-shot accuracy on 6 tasks with 250 samples each from the eval set. These tasks include image emotion recognition, video emotion recognition, persuasion strategy classification, MSRVTT, HVU and MSVD-QA

| Model | Audio Summarization (3 Shot) | | | IMDb Sentiment | |
|---|---|---|---|---|---|
|  | BLEU | ROUGE | METEOR | 0-shot | 1-shot |
| Behaviour-LLaVA | 19.0 | 25.1 | 39.3 | 84.1 | 90.2 |
| LLaMA-VID | 15.1 | 18.3 | 30.7 | 80.3 | 87.9 |
| VALOR (Chen et al., 2023a) | 6.6 | 10.0 | 23.9 | - | - |
| **Improvement of Behavior-LLaVA over LLaMA-Vid** | 25% | 37% | 28.01% | 4.73% | 2.61% |

Table 12: Evaluation on audio and text modalities. We evaluate on the audio summarization benchmark(Han et al., 2023) for audio and IMDB sentiment benchmark for text.(Maas et al., 2011)

| Task | 0-Shot improvement over Llama-Vid |
|------|-----------------------------------|
| LVU | 21.49% |
| Video Ad Understanding | 43.18% |
| Video Emotion | 51.85% |
| Image and Video Memorability | 186.4% |
| Video QA | 0.6% |
| Image Emotion | 29.14% |
| Image Dense Captioning | 4.95% |
| HVU | 5.88% |
| Audio Summarization | 30% |
| Sentiment Analysis | 4.73% |

Table 13: Average 0-Shot performance improvement of Behaviour-LLaVA over Llama-Vid across all tasks including Image, Video, Audio, and Language modalities. The table shows that Behaviour-LLaVA improves over its base model on all tasks.

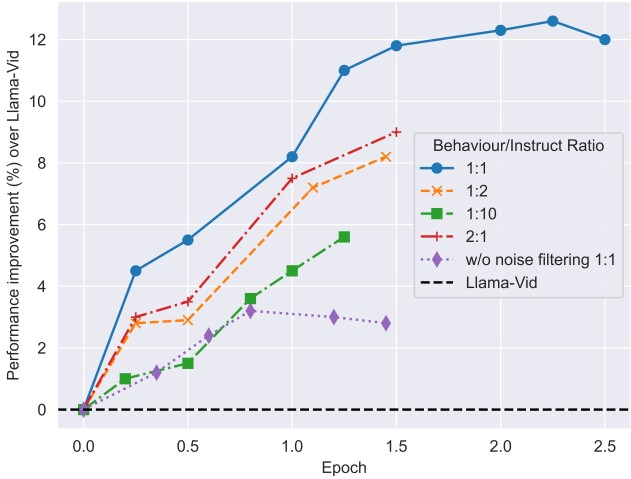

Figure 18: Percentage performance improvement over an untrained LLaMA-Vid model, compared across various sampling ratios at different checkpoints. The 1:1 sampling ratio shows the best empirical performance. Performance is averaged over 0-shot accuracy improvements on six tasks with 250 samples each from the evaluation set. These tasks include image emotion recognition, video emotion recognition, and persuasion strategy classification, MSRVTT, HVU and MSVD-QA. The figure also shows the benefit of our data filtering process. While training on unfiltered BLIFT also improves the result over baseline performance of Llama-Vid but data filtering adds more improvement on top of it.

## C  LISTINGS

### Listing 2: GPT-4V Prompt to calculate correctness of a image dense caption

```
You are a great  critique  for  analyzing  images  and  captions .

Assess  the  performance  of  a  dense  image  captioning  model  based  on  the  correctness  of  the  captions  generated .

Please  assess  the  correctness  of  the  provided  caption  in  relation  to  the  image.  Consider  whether  the  caption  accurately
      identifies  and  describes  the  main  subjects  or  objects  depicted  in  the  image.  Assess  whether  the  caption  correctly
      interprets  the  relationships  between  elements  within  the  image,  such  as  actions ,  interactions ,  or  spatial  arrangements.
      Focus  on  the  precision  and  accuracy  of  the  information  presented  in  the  caption .  Provide  a  score  reflecting  the  level
      of  correctness ,  ranging  from  1  (low  correctness )  to  10  (high  correctness ).
```

### Listing 3: GPT-4V Prompt to calculate detail of a image dense caption

```
You are  a  great  critique  for  analyzing  images  and  captions .

Assess  the  performance  of  a  dense  image  captioning  model  based  on  the  detail  of  the  captions  generated .
```

Evaluate the level of detail captured in the provided caption. Consider how well the caption describes specific attributes, features, or aspects of the image, including colors, shapes, textures, sizes, and any other relevant details. Assess whether the caption provides comprehensive information about the scene depicted in the image, covering both prominent and subtle elements. Pay attention to the depth and specificity of the details conveyed in the caption. Provide a score indicating the richness of detail, ranging from 1 (low detail) to 10 (high detail).

Listing 4: GPT-4V Prompt to calculate quality of a image dense caption

You are a great critique for analyzing images and captions.

Assess the performance of a dense image captioning model based on the quality of the captions generated.

Assess whether the caption is concise yet descriptive, providing meaningful and engaging information about the image. Evaluate the caption's ability to evoke a clear mental image corresponding to the visual content. Additionally, consider if the caption is insightful or imaginative in its description. Provide a score reflecting the overall quality of the caption, ranging from 1 (low quality) to 10 (high quality).

Listing 5: Prompt to generate verbalization using Llava-1.6

USER: For the given image, write a one line caption and maximum 20 descriptive keywords, no more than that.
For example:
{"caption": "This is a sample caption", "keywords": "keyword_1, keyword_2, keyword_3"}
Answer in JSON format only. Do not include any other information in your answer.
⟨image⟩
ASSISTANT:

Listing 6: Perceptual Signal Instruction fine-tuning template for the image: http://farm6.staticflickr.com/5106/5670500150_e035dd2d30_z.jpg

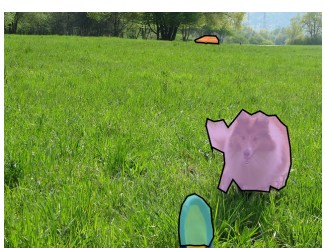

⟨SYSTEM⟩You are an AI visual assistant. Answer all questions as you are seeing the media⟨SYSTEM⟩⟨USER⟩The objects in this image in no particular order are car, dog, frisbee. Give me the order of saliency of these objects, start with the most salient object and end with the least salient object, each in a separate line. Give me the objects only and nothing else.
⟨image⟩
⟨ASSISTANT⟩
dog
frisbee
car
⟨ASSISTANT⟩

Listing 7: Perceptual Signal Instruction fine-tuning template for the image: http://farm6.staticflickr.com/5106/5670500150_e035dd2d30_z.jpg

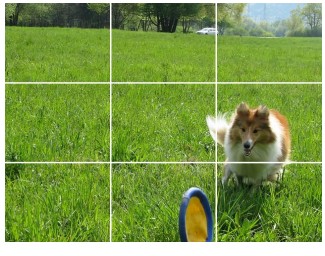

⟨SYSTEM⟩You are an AI visual assistant. Answer all questions as you are seeing the media⟨SYSTEM⟩⟨USER⟩Assume the given image is broken into a 3X3 grid the regions or tiles being named "upper−left" "upper−center", "upper−right", "middle−left", "middle−center", "middle−right", "bottom−left", "bottom−center", "bottom−right". Rank these regions or tiles based on their saliency, give me the line separated ranking of all regions in decreasing order.
⟨ASSISTANT⟩
middle−right
bottom−center
bottom−right
upper−center
upper−right
middle−center
upper−left
middle−left
bottom−left

Listing 8: Ad-LLaVA Instruction fine-tuning template for the video: https://www.youtube.com/watch?v=BKPQkjRF4yY

```
<SYSTEM>You are an AI visual assistant. You are given a detailed description of a media, followed by the actual media. Answer all
    questions as if you are seeing the media.</SYSTEM>

<USER>
The video advertisement is titled "Gatorade | Make Your Own Footsteps with Suni Lee" for the brand Gatorade. The audio in the ad
    says "[ASR HERE ...]". Analyze this video deeply, then write scene by scene description of the video and answer the following
    questions. What percentage of viewers would like this video, and what would be the top-5 popular comments on this video?
    What would the replay graph values for each scene be?
<video>...</video>
</USER>
```

```
<ASSISTANT>
The scene-by-scene descriptions are:

Scene 1: The scene shows a woman looking off into the distance with an orange line going around the outside of the screen. The
        foreground colors of the scene are black, mud green, gray, dark gray, and the background colors are dark brown, black, dark
        gray. The dominant tone of the scene is neutral. This scene is categorized by the tags: cosmetic, eyebrow, face, girl,
        ponytail, stand, string, woman.
Scene 2: The scene shows a woman balancing on a skateboard in a yard. The foreground colors of the scene are black, mud green, dark
        gray, olive, and the background colors are black, dark gray, gray, dark brown.The dominant tone of the scene is neutral. This
         scene is categorized by the tags: athletic, balance, beam, car, girl, house exterior, hurdle, jog, legging, plank, rail,
        seesaw, woman, yard.
Scene 3: The scene shows a girl jumping over a wooden ramp in the backyard. The foreground colors of the scene are black, dark gray,
        gray, dark blue, and the background colors are dark brown, dark blue, purple, dark pink, brown.
...
</ASSISTANT>
```

## D  DATASET DESCRIPTIONS

1. MSVD-QA and MSRVTT-QA: These datasets are based on Microsoft Research Video Description (Chen & Dolan, 2011) and MSR-VTT corpora (Xu et al., 2016b) and are extensively used in many video captioning and question-answering experiments. The MSVD-QA dataset has a total number of 1,970 video clips and 50,505 question-answer pairs. The MSRVTT-QA dataset contains 10K video clips and 243k question-answer pairs.
2. ActivityNet-QA (Caba Heilbron et al., 2015) is a benchmark primarily for human activity understanding containing 849 hours of video, including 28,000 action instances.
3. VideoEmotion-8 (Asur & Huberman, 2010) dataset comprises 1,101 user-generated videos sourced from YouTube and Flickr, each containing a minimum of 100 videos per emotional category, as per Plutchik Wheel's emotion model.
4. Ekman-6 (Xu et al., 2016a) dataset is compiled from social websites, with each of its 1,637 videos labeled with a single emotion category based on Ekman's psychological research.
5. CAER (Lee et al., 2019) dataset, sourced from TV shows, consists of 13,201 clips with an average sequence length of 90, each manually labeled with six basic emotions, aligning with the Ekman-6 dataset.
6. IAPSa (Mikels et al., 2005) is a subset of IAPS, following the Mikels model with eight emotion categories such as amusement, awe, contentment, excitement, anger, disgust, fear, and sadness. It consists of 395 affective images, marking the first visual emotion dataset with discrete categories.
7. Emotion6 (Peng et al., 2015) features 1,980 images sourced from Flickr, each labeled by 15 annotators according to the Ekman model, covering six emotion categories: happiness, anger, disgust, fear, sadness, and surprise.
8. EmoSet (Yang et al., 2023) encompasses a total of 3.3 million images, including 118,102 from social networks and artistic sources, evenly distributed across various emotion categories. Based on the Mikels model, EmoSet is categorized into eight emotion categories.
9. Abstract (Machajdik & Hanbury, 2010) exclusively consists of color and texture combinations without recognizable objects. Differing from the IAPS dataset where emotions often stem from identifiable objects, the abstract paintings dataset was peer-rated via a web survey, with each image rated approximately 14 times. It comprises 228 images spanning eight categories similar to those in IAPS.
10. Memento10k(Newman et al., 2020) is a short-term video memorability dataset comprising 10,000 video clips, with 900,000 human memory annotations recorded at various delay intervals. The video clips were, on average, 3s long.
11. VideoMem (Cohendet et al., 2019) is comprised of 10,000 soundless videos, each lasting 7 seconds, accompanied by memorability scores. Memorability was measured twice: first, shortly after viewing and again 24-72 hours later to capture both short-term and long-term memorability effects.
12. LaMem (Khosla et al., 2015) dataset is a short-term image memorability dataset comprising of 60000 images. The dataset contains scene-centric images, object-centric images and other types such as images of art, images evoking certain emotions, and other user-generated images.
13. SUN (Isola et al., 2011) dataset is a short-term image memorability dataset comprising 2222 images that were sourced from the SUN database.
14. MemCat (Goetschalckx & Wagemans, 2019) dataset is a short-term image memorability dataset comprising 10000 images. It consists of five broader memorability-relevant semantic

categories (animal, sports, food, landscapes, vehicles), with 2K exemplars each, further divided into different subcategories (e.g., bear, pigeon, cat, etc. for animal). The images were sourced from existing image sets: ImageNet, COCO, Open Images Dataset, and SUN.

15. MediaEval (Kiziltepe et al., 2021) utilized publicly available links to short-form video clips, each averaging 6 seconds in duration, with both short-term and long-term memorability scores. Short-term memorability evaluations were conducted on videos viewed within the preceding few minutes, while long-term memorability assessments were based on videos viewed within the previous 24 to 72 hours.

16. LAMBDA (I et al., 2024) is a long-term memorability consisting of 2205 video ads collected over 1749 participants covering 276 brands. The average video length is 33 seconds and the videos are highly complex, consisting of audio, logos, fast-moving scenes, emotions, *etc*.

# E  HYPERPARAMETERS

## E.1  TRAINING HYPERPARAMETERS

1. Deepspeed Zero2 with Offload
2. Base Model: llama-vid-13b-full-224-video-fps-1
3. Version: imgsp_v1
4. Vision Tower: LAVIS/eva_vit_g.pth
5. Image Processor: processor/clip-patch14-224
6. Multimodal Projector Type: mlp2x_gelu
7. Multimodal Vision Select Layer: -2
8. Disable Use of Start/End Tokens for Images
9. Disable Use of Patch Tokens for Images
10. Image Aspect Ratio: Pad
11. Video Token: 2
12. BERT Type: qformer_pretrain_freeze_all
13. Number of Queries: 32
14. Compression Type: Mean
15. Enable bf16 Precision
16. Number of Training Epochs: 2.2
17. Per-Device Training Batch Size: 4
18. Per-Device Evaluation Batch Size: 4
19. Learning Rate: 2e-5
20. Weight Decay: 0
21. Warmup Ratio: 0.03
22. Learning Rate Scheduler: Cosine
23. Maximum Sequence Length: 2048
24. Enable Gradient Checkpointing

## E.2  VERBALIZATION MODULE

1. Scene Splitting: pyscenedetect (Breakthrough, 2023)
2. ASR
   (a) openai/whisper-large-v3
   (b) Batch Size: 200
   (c) Chunk Length (s): 30
3. Caption and Keywords: llava-v1.6-vicuna-13b 5

# F  LIMITATIONS

In this paper, we try to show the hypothesis that training on the behavior modality improves learning of the content modality. We train the models on comments and likes to show this. We test our models on multiple benchmarks and obtain positive results. While we try to cover a wide variety of tasks and while results do conclusively show that the hypothesis is true, yet, we can test on more benchmarks covering even more tasks. Similarly, we can show it with multiple models, other than LLaMA-Vid.

## G    BROADER IMPACTS

Our paper talks about how behavioral training can positively impact content understanding of VLMs. We think this will be useful in various content understanding applications such as question answering, captioning, etc.

## H    ETHICAL IMPLICATIONS

This paper demonstrates that training on behavioral modalities enhances the learning of content modalities. Models trained on user interactions such as comments and likes were tested on multiple benchmarks and yielded positive results. While these findings present exciting opportunities for advancing content understanding in AI systems, they also raise important ethical considerations that must be carefully addressed.

1. No personally identifiable information (PII) is used to improve content understanding. Instead, aggregated behavioral data, including replays, likes, and comments, is utilized, ensuring user privacy. We have implemented rigorous anonymization and aggregation techniques to protect individual user identities.

2. We explicitly acknowledge the inherent biases that may exist in data sourced from social media platforms such as Reddit and YouTube. Despite our rigorous data filtering and cleaning processes, we recognize that the dataset may still be subject to demographic skews, self-selection bias, and algorithmic influences. These biases could potentially lead to uneven model performance across different user groups or reinforce existing societal biases. To mitigate these issues, we emphasize the importance of considering these broader implications when applying our model and interpreting its results.

3. We acknowledge the need for greater cultural diversity in our dataset. To address this, we plan to release our artifacts as open-source and encourage community contributions to incorporate multilingual and multicultural data. This could involve expanding the range of subreddits and YouTube channels included in our dataset, with the aim of capturing a more diverse and representative spectrum of receiver behavior across different cultural contexts. By taking this approach, we hope to enhance the ethical considerations and societal impact of our work, providing a more holistic view of behavioral patterns in various cultural settings. However, we also recognize that this approach may introduce new challenges, such as ensuring the quality and reliability of community-contributed data.

4. To ensure the safety and ethical integrity of our dataset, we implemented a comprehensive content filtering and monitoring strategy. During data preparation, we employed advanced filtering mechanisms including LLaMA-Guard-3-1b for content safety assessment, systematically removing posts, comments, and media flagged with NSFW tags from community-moderated platforms. We applied an extensive bad word list to screen potentially offensive content and retained only data explicitly marked as safe for use. The filtering process involved multi-stage validation to identify and eliminate potentially harmful or biased content, focusing on maintaining the dataset's quality and ethical standards. Our generated responses underwent rigorous safety checks, utilizing automated moderation tools and community-developed safety tags to minimize the risk of generating inappropriate or harmful content. By combining automated filtering, community-sourced moderation tags, and machine learning-based safety assessments, we developed a robust approach to content monitoring that prioritizes responsible AI development and user protection.

