# OpenReview forum: "Teaching Human Behavior Improves Content Understanding Abilities Of VLMs"
_ICLR.cc/2025/Conference — ICLR 2025 Poster_

### Official Review · Reviewer_Kqw1 · 2024-10-31

**Soundness:** 2
**Presentation:** 3
**Contribution:** 3
**Rating:** 6
**Confidence:** 4

**Summary:**

1. This paper proposes the use of behavioral data training to enhance the content understanding capabilities of multimodal models and has collected a cross-modal dataset called BLIFT.

2. Building on LLaMA-Vid, this paper applies two different instruction fine-tuning strategies, Behavior-LLaVA and Ad-LLaVA, on the BLIFT dataset. The Behavior-LLaVA strategy leverages behavioral data, making it stronger than the Ad-LLaVA strategy, and demonstrates superior performance compared to baseline models across multiple multimodal downstream tasks.

**Strengths:**

1. The paper presents an innovative perspective by examining large models' understanding of data through the lens of human behavioral performance.

2. The authors have compiled a substantial multimodal dataset and fine-tuned the model from the perspective of behavioral performance, demonstrating impressive results in downstream tasks.

**Weaknesses:**

1. Lacks sufficient information on data collection, processing, and distribution. The description of the data collection and filtering process lacks logical coherence. The filtering appears to be largely empirical, and there is no thorough analysis of the feature distribution of the collected data.

2. The paper fails to provide information about the specific experimental details used for instruction tuning (including in the appendix) and does not compare the changes in time efficiency resulting from tuning with additional commentary data versus without.

3. The paper states that sensor data is inferior to behavioral data based on the insufficient quantity of sensor data. It could be argued that sensor data is challenging to collect; however, comparing the quality of sensor data requires more substantial evidence.

4. In this paper, "Ad-LLaVa" sometimes appears as "AdLLaVa." Please ensure consistency in spelling.

**Questions:**

1. Did the authors consider the issue of distributional bias during data collection, and how might this impact the downstream tasks? The extent of data acquisition is also an important aspect of quality assessment. While collecting large-scale data, it's essential to consider the characteristics of the data being collected. Besides YouTube or Reddit, have other data sources been considered? In addition, please address the legality of your data acquisition and processing.

2. Please provide more experimental details, including experimental design and setup. Additionally, only the source code for llama-vid is provided—have any model improvements beyond fine-tuning been considered? The supplementary materials only include the llama-vid code, with the only visible data being a small amount of Reddit data.

---

> ### Author Response · Authors · 2024-11-20
> **Thank you for your review!**
>
> We thank the reviewer for taking the time and effort to review our paper.
>
>
> >Data collection and processing,  description of filtering process
>
> We have tried our best to explain the dataset curation in Section 2.1 by providing detailed instructions to reproduce the collection, filtering, and processing behavior data to BLIFTs and added a figure to easily visualize the same (Fig 12 in the new version).
> The raw data was collected from two subreddits, r/pics and r/videos. The initial filtering excluded posts from before February 2015 in r/pics, following a rule banning overlay text, and restricted the dataset to posts from January 2018 onwards. Refinement steps removed NSFW, BOT-generated, deleted, and duplicate content. Comments with fewer than three words were filtered, and deduplication was performed using a TF-IDF similarity threshold of 0.6. Posts with fewer than two comments and videos longer than 500 seconds were excluded, resulting in 631,000 images and 397,000 videos. Further, we limited the dataset to Reddit-hosted images and YouTube videos, excluding 51% of YouTube content due to privacy issues, leaving a final dataset of 400,000 images, 80,000 videos, and over 1.8 million comments.Once the data is collected, we verbalize the collected images and videos. For images, we generate captions and keywords and for videos we first split the video to scenes, generate caption and keywords for the individual scenes and generate ASR for the video. We have attached the code for these and comments deduplication in the supplementary of the current version and describe the process below.
> We also show a multitude of ways of creating multimodal instruction tuning from behavior data (as also kindly pointed out by Reviewer 6UPg). To make it even more comprehensive, we have added the verbalization details Section D.2 of the appendix. Details of the verbalization module (Also added to section D.2)
>
>
> - Details of the verbalization module (Also added to section D.2)
>   1. Scene Splitting: pyscenedetect (Breakthrough, 2023)
>
>   2. ASR
>     - openai/whisper-large-v3 with batch size of 200 and chunk length (s) of 30 seconds.
>
>   3. Caption and Keywords:
>     - Model: llava-v1.6-vicuna-13b 5
>     - Prompt for generating verbalization using LLaVA (added to supplementary - Listing-5 in the new version)
>
> - To present the raw data better, we have added a few qualitative samples using actual Reddit threads and YouTube comment sections from our data (Fig 13 and 14) in the new version. The figures show that content and their associated behavior provide insights into various elements, including style, topic, emotional content, conceptual themes, object identification, aesthetic qualities, and descriptive details.
>
>
> > Analysis of the feature distribution of the collected data, issue of distributional bias
>
> Thank you for the suggestion. We have included a dataset analysis covering various aspects of the collected data in Section A- Dataset Analysis in the new version. These figures show that the BLIFT dataset covers complex and diverse types of images, videos, and responses, including emotions, art styles, objects, background/foreground scenes, concepts, and themes. We describe the setup below.
> - **Topic Analysis:** To understand the topic distribution of the images and videos in the BLIFT dataset, we extracted the topics from titles, captions, and keywords for images and titles, descriptions, scene captions and keywords for videos using BERTopic. These topics were clustered and assigned a name by GPT-4o-mini  and visualized using a sunburst graph (Figures 15 and 17 in the new version of the paper).
> Comments: We similarly analyzed the topic distribution of receiver feedback for both images and videos (Figure 16 and 18)
>
> - **Qualitative Samples and Comments**: Qualitative examples of posts and comments from both reddit and YouTube are provided in Figures 13 and 14.  The figures show that content and their associated behavior provide insights into various elements, including style, topic, emotional content, conceptual themes, object identification, aesthetic qualities, and descriptive details.
>
> We provide more details of the data below:
> - Average Length of Verbalization: Videos: 2,107 tokens, Images: 321 tokens
> - Average Number of Comments: Videos: 3.1 comments, Images: 3.8 comments
> - Number of Scenes:
>   - Reddit: Mean = 24, Std = 17
>   - YouTube: Mean = 13, Std = 5
> - Likes-to-Views Ratio Distribution: Mean = 0.022, Std = 0.003

---

> ### Author Response · Authors · 2024-11-20
>
> > The filtering appears to be largely empirical
>
> To quantify the impact of our filtering steps on the final performance, we have compared the performance from training on unfiltered data with filtered data (Figure 11 in the new version). The figure shows the benefit of our data filtering process. While training on unfiltered BLIFT also improves the result over the baseline performance of Llama-Vid, but data filtering adds more improvement on top of it. It can be seen from the figure that unfiltered Behavior-LLaVA only has a 2.75% average improvement in performance. On the other hand, filtered Behavior-LLaVA has a 12% improvement.
>
>
> > specific experimental details, including experimental design and setup Additionally, only the source code for llama-vid is provided
>
> We provide details about the instruction tuning template in Figure 4. The specific hyperparameters used are documented in Appendix Section D.1 in the new version
> - ⁠Deepspeed: Zero2 with Offload
> - Base Model: llama-vid-13b-full-224-video-fps-1
> - Version: imgsp_v1
> - Vision Tower: LAVIS/eva_vit_g.pth
> - Image Processor: processor/clip-patch14-224
> - Multimodal Projector Type: mlp2x_gelu
> - Multimodal Vision Select Layer: -2
> - Use of Start/End Tokens for Images: Disabled
> - Use of Patch Tokens for Images: Disabled
> - Image Aspect Ratio: Pad
> - Video Token: 2
> - BERT Type: qformer_pretrain_freeze_all
> - Number of Queries: 32
> - Compression Type: Mean
> - bf16 Precision: Enabled
> - Number of Training Epochs: 2.2
> - Per-Device Training Batch Size: 4
> - Per-Device Evaluation Batch Size: 4
> - Learning Rate: 2e-5
> - Weight Decay: 0
> - Warmup Ratio: 0.03
> - Learning Rate Scheduler: Cosine
> - Maximum Sequence Length: 2048
> - Gradient Checkpointing: Enabled
>
>
> > Time Efficiency
>
> Thank you for the suggestion. We have benchmarked the time efficiency of BehaviorLLaVA (with behavior data) and AdLLaVA (without behavior data). In the interest of reproducibility, we have provided all the hyperparameters (refer to section D.1), and  the training and inference code were provided in the supplementary.
> - Experimental Setup: We train all our models on 2 nodes of 8xA100 GPUs. We use gradient checkpointing, bf16, and deepspeed zero2 default config.
> - We first compare the times taken to finetune on BLIFT (Behavior-LLaVA) and BLIFT without behavior data (Ad-LLaVA) which take 23.5 and 19 hours respectively.
> - Inference Comparison: Since both models use the same instruction format for inference they take similar times. Therefore Behavior-LLaVA takes ~20% more time to train but there is no change in efficiency during inference.
>
>
> > It could be argued that sensor data is challenging to collect
>
> We completely agree with the reviewer that the reason for the finding that sensor data empirically performs inferior to behavioral data could be that the amount of sensor data available is lesser than behavioral data. We have reasoned on a similar line in Lines 90-96. In general, as you have rightly indicated, sensor data or perceptual signals are usually collected in lab settings and are more expensive to collect and, therefore, are not available at scale.
>
>
> > only visible data being a small amount of Reddit data
>
> We will release the complete BLIFT upon acceptance.
>
>
> > "Ad-LLaVa" sometimes appears as "AdLLaVa."
>
> Thank you for pointing this out. We will ensure that "Ad-LLaVa" is used consistently throughout the paper and make the necessary corrections.

---

> ### Author Response · Authors · 2024-11-20
>
> **Regarding the compliance of the release of Reddit threads and YouTube comments and likes with the platforms’ terms of use:**
>
> We have covered the details in the ethics section of the paper (Section F). We further expand on terms of use below:
>
> We plan to release the following artifacts:
> - YouTube video IDs, comment IDs, titles, descriptions, views, likes, replay graphs, dates, channel names, speech transcripts, and scene verbalization along with timestamps
> - Reddit post IDs, comment IDs, upvotes, deleted, NSFW, bot flag, timestamp, media links, post title, number of comments on the post, automatically generated captions and keywords of the media files.
>
> The dataset does not contain any Personally Identifiable Information. All of the released information is aggregated and anonymized. We plan to release the IDs (and not the raw bytes) of the videos, YouTube comments, Reddit posts, and comments. This practice has been followed in past research works. We have given a few examples of such past works at the end.
>
> **On other data sources**
> We limited our focus to YouTube and Reddit because they provide the most diverse and globalized content and receiver feedback at scale. Further, both these sources have been utilized in previous literature and training of LLMs and VLMs in the past (Gemini, ChatGPT, FLAVA, MineDojo, OPT-IML) due to their permissive licenses and ready availability and applicability.
>
> **Past works using Reddit and YouTube**
>
> - YouTube
>     - Mine dojo (NeurIPS 2022 Outstanding Award Paper)
>       - https://openreview.net/forum?id=rc8o_j8I8PX
>       - 730k YouTube Videos with description and titles, 340k Reddit posts and upvotes
>     - VidChapters-7M (NeurIPS 2023)
>       - https://antoyang.github.io/vidchapters.html
>       - 7 million YouTube videos with their chapters, speech transcripts, and chapter annotations
>     - LCBM (ICLR 2024)
>       - ​​https://openreview.net/forum?id=TrKq4Wlwcz
>       - 25k YouTube videos with likes, views, titles, descriptions, dates, channel names, speech transcripts, and scene verbalizations
>     - Social Media Attributions in the Context of Water Crisis (EMNLP 2020)
>       - https://aclanthology.org/2020.emnlp-main.109.pdf
>       - 70k comments, 43k users
>     - VTC (ECCV 22)
>       - https://unitaryai.github.io/vtc-paper/
>       - 30k videos, comments Youtube + Reddit comments
>
> - Reddit:
>     - RedCaps (Neurips 2021)
>       - https://arxiv.org/pdf/2111.11431 https://redcaps.xyz
>       - 12M pairs, 13 years, 350 subs, licensed under CC-4.0
>     - OPT-IML dataset extensively used in LLM fine-tuning (cited 240 times) is built on Reddit threads
>       - Link: https://arxiv.org/abs/2212.12017
>     - Mine dojo (Neurips 2022 Outstanding Award Paper)
>       - https://openreview.net/forum?id=rc8o_j8I8PX
>       - 730k Youtube Videos, 340k Reddit posts
>     - The papers you have also kindly indicated to us release Reddit comments and upvotes
>       - Dialogue Response Ranking Training with Large-Scale Human Feedback Data (https://github.com/golsun/DialogRPT?tab=readme-ov-file) released under MIT license
>       - How to be Helpful on Online Support Forums? (https://aclanthology.org/2022.wnu-1.3/)
>     - Tl; dr: Mining Reddit to learn automatic summarization (ACL Workshop, cited 275 times) (https://aclanthology.org/W17-4508/)
>     - Instagram Influencers Dataset (WSDM 21):
>       - https://sites.google.com/site/sbkimcv/dataset/instagram-influencer-dataset?authuser=0
>       - 10M instagram posts, likes, comments

---

### Official Review · Reviewer_jLqC · 2024-10-31

**Soundness:** 3
**Presentation:** 2
**Contribution:** 2
**Rating:** 5
**Confidence:** 3

**Summary:**

This paper demonstrates hat training VLMs to predict receiver behaviors, such as likes, comments, and replay graphs, which are available at scale, enhances the VLM’s performance across a broad range of downstream content understanding tasks. Next, this paper constructs a behavior-LLaVA instruction fine-tuning dataset, and obtain the fine-tuned Behavior-LLaVA and AdLLaVA. The experiments on the fine-tuned VLMs demonstrate the effectiveness of content and behaviour, and study the different effect of perception and action behavior.

**Strengths:**

1. The constructed dataset could be very helpful for VLM researches.
2. The fine-tuned VLMs show improvement across various tasks.

**Weaknesses:**

1. The idea of the connection between VLM performances and fine-tuning on behavior data, upon which the paper is built, is questionable. The authors try to draw some analogy from relevant studies in human behavior, but fail to make a clear point with the work done in this paper (fine-tuning VLMs on behavior datasets). Also, the logic in this paper lacks clear or formal theoretical analysis on why fine-tuning VLMs on behavior datasets could improve performance, but rather just some abstract references to some concepts in the human behavior research. In this respect, the logic foundation of this paper is questionable and unconvincing.
2. The study of the connection between LLM/VLM foundation models and human behaviors is not novel. In fact, many researches [1][2][3] have been performed from various different perspectives, such as social behavior and trust behaviors. If we are to view the study on the connection of behavior with VLMs as a contribution in this paper, there are already many researches delivering similar and more comprehensive conclusions on this topic. In this sense, the authors should clearly position this paper along the research line and clearly state the difference between other related works.
3. The conclusion that learning behavior leading to learning content better is quite trivial and might not be very useful. In fact, adding more fine-tuning data of any types could lead to performance improvement on some relevant tasks. If the authors really want to make this point, they should curate the same dataset without explicitly telling VLMs what the behavior is and explore the performance of the VLM fine-tuned on this dataset.

References:

[1] Xie, Chengxing, et al. "Can Large Language Model Agents Simulate Human Trust Behaviors?." arXiv preprint arXiv:2402.04559 (2024).

[2] Li, Yuan, Yixuan Zhang, and Lichao Sun. "Metaagents: Simulating interactions of human behaviors for llm-based task-oriented coordination via collaborative generative agents." arXiv preprint arXiv:2310.06500 (2023).

[3] Chen, Weize, et al. "Agentverse: Facilitating multi-agent collaboration and exploring emergent behaviors." The Twelfth International Conference on Learning Representations. 2023.

**Questions:**

Any analysis on the harmfulness/trustworthiness/bias on the generated content from behavior LLaVa? The collected dataset could contain these harmful content.

---

> ### Author Response · Authors · 2024-11-20
> **Thank you for your review!**
>
> We thank the reviewer for taking the time to review our paper. Below, we try to address the reviewer's questions and concerns.
>
>
> > why fine-tuning VLMs on behavior datasets could improve performance, unclear idea of the connection between VLM performances and fine-tuning on behavior data,
>
> Human behavior occurs as a downstream artifact in the process of communication. Behavior (such as likes, shares, upvotes, comments, etc) is produced by a receiver as a response to the message sent by the sender. Being a downstream effect, behavior can help us infer important signals about the message itself. For example, a person’s rising heartbeat upon watching the action scenes in the movie Jurassic Park can tell us about the excitement levels in the movie. Similarly, regressing (gonig back and reading again) while reading a text is indicative of important or confusing phrases in the text. Our hypothesis is that these behavioral signals (like heart beat, regression, etc. in these examples), if properly harnessed, should be able to increase performance on the message understanding tasks popular in NLP and CV, like question answering, sentiment analysis, topic classification, etc. Despite this, behavior data is considered noise and is ignored while training large language models and also large vision and language models. For instance, Pile is a popular dataset used for training LLMs. (Biderman 2022) mention that they remove all metadata (containing behavioral data) while creating Pile and subsequently training Pythia. Similarly, behavioral data was removed while creating LLaVA-Instruct-150k (Liu et al 2023), MiniGPT (Zhu et al 2023), and RefinedWeb (Penedo et al 2023).
>
>
> > proposed method might not be very useful
>
> Through the proposed method, we achieve improvements over the base VLM across six types of behavior, spanning 46 tasks across 26 benchmark datasets in both zero-shot and fine-tuned settings. These gains are demonstrated on a wide range of tasks, from low-level content understanding (e.g., object and activity recognition) to high-level tasks (e.g., topic and emotion detection). Additionally, our method delivers superior or competitive performance compared to state-of-the-art domain-specific models, 2.5x larger VLMs, and even proprietary models like GPT-4V. Our approach introduces a scalable way to enhance the content understanding capabilities of VLMs, requiring minimal cost and no architectural changes. We sincerely value your feedback and aim to align on key conclusions. If there are specific analyses or experiments that the reviewer can point out that would address any remaining concerns, we are fully committed to performing them during the rebuttal phase to reach a shared conclusion.
>
>
> >curate the same dataset without explicitly telling VLMs what the behavior is and explore the performance of the VLM fine-tuned on this dataset.
>
> There may have been some confusion in our presentation, but this is exactly why we train Ad-LLaVA. We train Ad-LLaVA on the BLIFT dataset without the behavior data, to disentangle the effect of learning behavior from learning content. We have added the Ad-LLaVA instruction tuning template in Listing 8 of the current version. A common trend we observe across all the experiments carried out is that Behavior-LLaVA performs better than the base model LLaMA-Vid and the finetuned model Ad-LLaVA on all tasks, especially in zero-shot settings.  In fact, Ad-LLaVA performs very similar to LLaMA-Vid itself. This shows that BLIFT adds meaningful signals on average (rather than noise) to the model. Interestingly, the performance gains remain even after fine-tuning on the task dataset (Tables-2,3,8).
>
>
> > Differences from [1,2,3], Proposed method is Trivial:
>
> Thank you for indicating these papers. These research papers try to simulate specific human behaviors of trust (Xie et al 2024) and coordination (Li et al 2023, Chen et al 2023). The focus of our paper is not on simulating specific types of human behavior, rather it is that teaching VLMs to simulate real behavior improves their content understanding capabilities. We show the results for this hypothesis on six types of action and perception behaviors (replays, memorability, comments, likes, upvotes, and saliency). We find that teaching behavioral data results in better performance across a wide variety of tasks and modalities. We have added these references and the differences between our work and these papers in the new version of the paper. To the best of our knowledge, the research area of improving content understanding using human behavior is largely unexplored, especially in VLMs and no works have shown improvements without any increase in parameters on such diverse range of tasks.

---

> ### Author Response · Authors · 2024-11-20
> **Analysis on the harmfulness/trustworthiness/bias on the generated content from behavior LLaVa**
>
> To mitigate potential risks of harmful or biased content when incorporating behavioral data from Reddit and YouTube, we implemented a comprehensive safety screening process for the dataset. Here are the steps we used filter our dataset (also expanded in Sections 2.1.1 and 2.1.2)
> Removed posts, comments, and media (by title) flagged with NSFW tags from community-moderated platforms
> Applied a comprehensive bad word list to screen potentially offensive content
> Utilized LLaMA-Guard-3-1b for additional content safety assessment and retained only content explicitly marked as safe for use
> Thanks for your suggestion, we also evaluate the generations of LLaMA-VID and BehaviourLLaVA. We generated dense captions and confirmed that safety rates measured through LLaMA-Guard-3-8b were consistent, we will add this to the Ethical Implications section.

---

> > ### Comment · Reviewer_jLqC · 2024-11-24
> >
> > Thanks for the author for providing the rebuttal! While I still think the conclusion on the connection between human behavior and VLM model performances still have some problems, I truly appreciate the efforts in curating a good dataset and conducting large-scale experiments. Therefore, I think this is a borderline paper with clear strengths and shortcomings. Maybe the ACs can decide whether the shortcomings with this research still warrant publication or not.

---

### Official Review · Reviewer_6UPg · 2024-11-03

**Soundness:** 4
**Presentation:** 2
**Contribution:** 3
**Rating:** 8
**Confidence:** 2

**Summary:**

Communication consists of not only the message but also the effects it has on the receiver. However, receiver behavior is commonly ignored when training vision language models. This paper shows that training VLMS on receiver behavior such as likes, comments and replay graphs improves their performance across 46 diverse tasks. Moreover, this data is commonly collected by default (through user interactions on Reddit and YouTube) and hence does not require additional annotation. They collect and release a large scale 730k-sample dataset containing this data.

**Strengths:**

1.	This proposed idea of using receiver feedback relating to images/videos is novel to the best of my knowledge (which should been taken with a pinch of salt since the reviewer mostly work on text-only alignment) and is certainly worthwhile exploring since the cost of using these receiver-behavior is essentially zero compared to how much it would cost for post-collection expert human annotations.
2.	The data collection and preprocessing from Reddit and YouTube are detailedly described and carefully thought-out. I’m especially impressed by the extensive data cleaning done by the authors as well as the multitude of ways that the content behavior data can be converted to instruction tuning data.
3.	The training setup designed by the authors seem reasonable, with well-designed ablations such as matching for additional data (Ad-LLaVA)
4.	The evaluation is done rigorously with 46 different tasks and general improvements across many of these metrics, in relation to strong baselines.

**Weaknesses:**

1.	The introduction claims that “behavior data is considered noise and is ignored while training large language models” (Line 50). However, this statement ignores the substantial work done in harnessing receiver-generated behavior data especially on online forums such as Reddit (through upvotes and comments), which have been used by many works including [1] and [2]. Nonetheless, I have not heard of VLMs using organic/behavioral feedback data so I would suggest the authors to acknowledge that prior work have been done for text-only settings and only claim that this is done in this paper for VLMs, which is more appropriate to the remainder of the paper.
2.	The organization for Section 3 Results and Discussion can be improved. Currently, it’s not easy to follow the main areas of improvement of the proposed methods. Because there are 46 evaluation metrics, it might be easier to follow if the results section starts with one table that shows the average gain in each task category e.g. VQA, which can be done by normalize all scores for a baseline model (e.g. LLaMa-Vid) to 100%, with the raw metrics in appendix. I would also recommend that the discussion is organized more clearly (i.e. each paragraph has one main point summarized as the paragraph heading) as it currently reads as simply describing the results and it’s hard for readers to follow along on the main insights arising from the results.

[1] https://arxiv.org/abs/2009.06978
[2] https://aclanthology.org/2022.wnu-1.3/

**Questions:**

To the best of my knowledge, platforms such as YouTube and Reddit have stringent Terms of Use requiring how such behavioral data can be used and released as a dataset. How did the authors work together with the platform to ensure that collecting and releasing this data does not breach their terms of use?

**Details Of Ethics Concerns:**

I am not sure if using/releasing a large set of Reddit threads and YouTube comments/likes will violate the terms of use for these services.

---

> ### Author Response · Authors · 2024-11-20
> **Thank you for your thoughtful review**
>
> Thank you for your thoughtful review and suggestions on how we can improve the paper. They are very helpful for improving the overall work.
>
> Please find below our responses to your comments.
>
> 1. “Behavior data is considered noise” - By this statement, we mean that behavioral data, such as likes, shares, upvotes, comments, etc, are filtered out while creating datasets for LLMs. For instance, Pile is a popular dataset used for training LLMs. Biderman et al. 2022 mention that they remove all metadata (containing Behavioral data) while creating Pile and subsequently training Pythia. Similarly, behavioral data was removed while creating LLaVA-Instruct-150k, MiniGPT, and RefinedWeb datasets. We have clarified this statement in the new version.  We agree with the reviewer that much work has been done in harnessing receiver behavior and using it in various forms. However, what we are trying to indicate here is that most of the work is in predicting the receiver behavior rather than using the receiver behavior to improve the content understanding capabilities of models. The two papers kindly indicated by you also try to predict behavior in the form of relevance (Gao et al 2020) and helpfulness (Wang et al 2022) prediction. We have added these and other references to try to cover this branch of literature better.
>
> 2. Organization of Section 3 Results and Discussion: We realized that Section 3 needs better readability. With this concern in mind, we had added Figure 3 in the submitted version, which tries to summarize the results for the main tasks. We think that your suggestion of adding a summary table is helpful. We have added the table given below as Table 14 in the current version of the paper. It covers a summary of all the tasks and the average gain in each task category. We will shift it to the main section in the camera-ready version. **Discussion section**: Thank you for the suggestion. As per your kind suggestion, we have arranged the discussion section so that each paragraph has one main point summarized as the paragraph heading.
>
>
> | **Task**                       | **0-Shot improvement over Llama-Vid** |
> |--------------------------------|------------------------|
> | **LVU**                        | 21.49%                 |
> | **Video Ad Understanding**     | 43.18%                 |
> | **Video Emotion**              | 51.85%                 |
> | **Image and Video Memorability** | 186.4%                |
> | **Video QA**                   | 0.6%                   |
> | **Image Emotion**              | 29.14%                 |
> | **Image Dense Captioning**     | 4.95%                  |
> | **HVU**                        | 5.88%                  |
> | **Audio Summarization**        | 30%                    |
> | **Sentiment Analysis**         | 4.73%                  |
> **Average 0-Shot performance improvement over Llama-Vid across various tasks**

---

> > ### Author Response · Authors · 2024-11-20
> >
> > **Regarding the compliance of the release of Reddit threads and YouTube comments and likes with the platforms’ terms of use:**
> > We have covered the details in the ethics section of the paper (Section F). We further expand on terms of use below:
> >
> > We plan to release the following artifacts:
> > - YouTube video IDs, comment IDs, titles, descriptions, views, likes, replay graphs, dates, channel names, speech transcripts, and scene verbalization along with timestamps
> > - Reddit post IDs, comment IDs, upvotes, deleted, NSFW, bot flag, timestamp, media links, post title, number of comments on the post, automatically generated captions and keywords of the media files
> >
> > The dataset does not contain any Personally Identifiable Information. All of the released information is aggregated and anonymized. We plan to release the IDs (and not the raw bytes) of the videos, YouTube comments, Reddit posts, and comments. This practice has been followed in past research works. We give a few examples of such past works here:
> >
> > - YouTube
> >     - Mine dojo (NeurIPS 2022 Outstanding Award Paper)
> >       - https://openreview.net/forum?id=rc8o_j8I8PX
> >       - 730k YouTube Videos with description and titles, 340k Reddit posts and upvotes
> >     - VidChapters-7M (NeurIPS 2023)
> >       - https://antoyang.github.io/vidchapters.html
> >       - 7 million YouTube videos with their chapters, speech transcripts, and chapter annotations
> >     - LCBM (ICLR 2024)
> >       - ​​https://openreview.net/forum?id=TrKq4Wlwcz
> >       - 25k YouTube videos with likes, views, titles, descriptions, dates, channel names, speech transcripts, and scene verbalizations
> >     - Social Media Attributions in the Context of Water Crisis (EMNLP 2020)
> >       - https://aclanthology.org/2020.emnlp-main.109.pdf
> >       - 70k comments, 43k users
> >     - VTC (ECCV 22)
> >       - https://unitaryai.github.io/vtc-paper/
> >       - 30k videos, comments Youtube + Reddit comments
> >
> > - Reddit:
> >     - RedCaps (Neurips 2021)
> >       - https://arxiv.org/pdf/2111.11431 https://redcaps.xyz
> >       - 12M pairs, 13 years, 350 subs, licensed under CC-4.0
> >     - OPT-IML dataset extensively used in LLM fine-tuning (cited 240 times) is built on Reddit threads
> >       - Link: https://arxiv.org/abs/2212.12017
> >     - Mine dojo (Neurips 2022 Outstanding Award Paper)
> >       - https://openreview.net/forum?id=rc8o_j8I8PX
> >       - 730k Youtube Videos, 340k Reddit posts
> >     - The papers you have also kindly indicated to us release Reddit comments and upvotes
> >       - Dialogue Response Ranking Training with Large-Scale Human Feedback Data (https://github.com/golsun/DialogRPT?tab=readme-ov-file) released under MIT license
> >       - How to be Helpful on Online Support Forums? (https://aclanthology.org/2022.wnu-1.3/)
> >     - Tl; dr: Mining Reddit to learn automatic summarization (ACL Workshop, cited 275 times) (https://aclanthology.org/W17-4508/)
> >     - Instagram Influencers Dataset (WSDM 21):
> >       - https://sites.google.com/site/sbkimcv/dataset/instagram-influencer-dataset?authuser=0
> >       - 10M instagram posts, likes, comments

---

> > > ### Comment · Reviewer_6UPg · 2024-11-24
> > > **Response to author comments**
> > >
> > > Thank you for addressing my concerns.
> > >
> > > 1. “Behavior data is considered noise”  -> thanks for your explanation. Minor note is that wherever relevant, it's better to frame this paper as a technique for improve VLMs (as it is in the pdf paper right now) - since the OpenReview page mostly say LLMs (e.g. in title; abstract), which is slightly confusing since the experiments are entirely about VLMs. It's ok to claim in the introduction that others have not done it even in the text-only context (given the explanation by the authors) but I think this paper will mostly be of interest to researchers working on vision LLMs rather than text-only LLMs.
> > >
> > > 2.  "Organization of Section 3 Results and Discussion" -> it does look clearer now, thanks for the change!
> > >
> > > 3. "Regarding the compliance of the release of Reddit threads and YouTube comments and likes with the platforms’ terms of use" --> I want to stress to the authors and others looking at my feedback in that I'm expressing genuine uncertainty in my comment - "I am not sure if using/releasing a large set of Reddit threads and YouTube comments/likes will violate the terms of use for these services." While I understand that there has been substantial past research that made use of and released materials from various social media data sources, my understanding is that the terms of use of these platforms were massively ramped around 2022 as monetization methods of these platforms changed when ChatGPT showed the world how valuable data is. Because I don't have a legal background, I want to raise this point out of an abundance of caution, especially to other legal specialists (who will presumably review this further). For the authors, it might be helpful to further clarify a. more details on how acess to youtube/reddit data was obtained (i.e. if there were some researcher access granted) and b. the proposed license for the BLIFT dataset they want to release.
> > >
> > > I will maintain my rating of 8 but I want to stress that this rating has a low confidence score (2) from a reviewer who is not familiar with vision-LLMs (but has experience with text-only LLM preference datasets).

---

### Official Review · Reviewer_VwGN · 2024-11-05

**Soundness:** 3
**Presentation:** 3
**Contribution:** 3
**Rating:** 8
**Confidence:** 3

**Summary:**

The paper proposes that using user behavior data collected from Reddit and YouTube helps large vision and language models achieve better performance across a wide range of language and visual content understanding tasks. The gains are especially more salient on the higher-level tasks such as emotion recognition, persuasion strategy classification, and question answering than the lower-level tasks like action and object
recognition. Furthermore, the ablation study demonstrates the importance of behavior data and instruction fine-tuning on behavior data.

**Strengths:**

1. The paper is well-motivated, highlighting the importance of leveraging user behavior data for content-understanding tasks.
2. It contributes a large-scale dataset BLIFT, consisting of 400k images and 330k videos, along with their receiver behavior, which is beneficial for research about vision language models.
3. The proposed training method has gained significant performance on a variety of vision-language tasks, which shows the effectiveness of the method.

**Weaknesses:**

More details of the training process can be provided for reproducibility, e.g. the average length of the text input, and how to deal with extremely long ASR results.

**Questions:**

NA

---

> ### Author Response · Authors · 2024-11-20
> **Thanks for your review**
>
> We sincerely thank the reviewer for their feedback.
>
> In the interest of reproducibility, we have added the code for verbalizing the images and videos (including scene detection, caption & keywords extraction, ASR extraction, and deduplicating comments). The training and inference code had already been provided in the supplementary materials at the time of submission. A sample dataset is also available in the supplementary materials, and we plan to release the complete dataset with the camera-ready version of the paper.
> Additionally, we have included details about the datasets, their sampling ratios, and the exact hyperparameters used for training in Section D for clarity. These details have also been summarized below for convenience.
>
> **Details**
> Training Details: Hyperparameters (Also added to Section C in appendix)
> - ⁠Deepspeed: Zero2 with Offload
> - Base Model: llama-vid-13b-full-224-video-fps-1
> - Version: imgsp_v1
> - Vision Tower: LAVIS/eva_vit_g.pth
> - Image Processor: processor/clip-patch14-224
> - Multimodal Projector Type: mlp2x_gelu
> - Multimodal Vision Select Layer: -2
> - Use of Start/End Tokens for Images: Disabled
> - Use of Patch Tokens for Images: Disabled
> - Image Aspect Ratio: Pad
> - Video Token: 2
> - Qformer Type: qformer_pretrain_freeze_all
> - Number of Queries: 32
> - Compression Type: Mean
> - bf16 Precision: Enabled
> - Number of Training Epochs: 2.2
> - Per-Device Training Batch Size: 4
> - Per-Device Evaluation Batch Size: 4
> - Learning Rate: 2e-5
> - Weight Decay: 0
> - Warmup Ratio: 0.03
> - Learning Rate Scheduler: Cosine
> - Maximum Sequence Length: 2048
> - Gradient Checkpointing: Enabled
>
> **Datasets used**
> - BLIFT (Ours)
> - [llava_v1_5_mix665k_with_video_chatgpt.json](https://huggingface.co/datasets/YanweiLi/LLaMA-VID-Data/blob/main/llava_v1_5_mix665k_with_video_chatgpt.json)
>
> **Dataset statistics**
> - Average length of the verbalization:
>    - Video: 2107 tokens
>    - Image: 321 tokens
> - Average number of comments:
>   - Videos: 3.1
>   - Images: 3.8
> - Number of scenes
>   - Reddit: mean = 24, std = 17
>   - YouTube: mean = 13, std = 5
> - Likes/Views ratio distribution
>   - Mean = 0.022, std=0.003
>
> **Topic analysis**
> To analyze the topic distribution across the BLIFT dataset, we employed BERTopic to extract topics from multiple elements: images, videos, and their associated comments, including titles, captions, and keywords. These topics were subsequently clustered and labeled using GPT-4o-mini. The resulting topic distribution and hierarchical relationships are visualized through sunburst charts in Figures 15-18. These visualizations demonstrate the breadth and diversity of topics and themes present in the BLIFT dataset's image, video, and comment content.
>
> **Qualitative Sample**
> We have added qualitative examples (fig 13 and 14) from BLIFT dataset, accompanied by their corresponding comments. These examples provide insights into various elements including style, topic, emotional content, conceptual themes, object identification, aesthetic qualities, and descriptive details.
>
> **Long ASR results**
> - Our videos are a maximum of 10-minutes long hence we are not handling ASR beyond the context length (2048 tokens)
> - We use 2 tokens (as opposed to 576 in llava) per frame to avoid context length issues.
> - ASR is embedded between Scene tokens following the LLaMA-VID LongVideoQA Format (as shown in Figure 3 of https://arxiv.org/pdf/2311.17043)

---

### Meta-Review · Area_Chair_QBYZ · 2024-12-18

**Metareview:**

This paper proposes leveraging large-scale user behavior data to enhance VLM performance on a broad spectrum of downstream tasks. The novel BLIFT dataset and the instruction-tuned Behavior-LLaVA and Ad-LLaVA models show promising gains, particularly on complex tasks like emotion recognition and persuasion strategy classification. Reviewers appreciate the paper's strong motivation, extensive data cleaning, and thorough evaluations across 46 tasks. Though some note that theoretical foundations and comparisons to related work in social and behavioral modeling could be more explicit, the consensus is that this approach is a meaningful contribution. The authors offer a new resource and a perspective that may inspire future work on integrating behavioral cues into VLM training. Given these strengths and the author's willingness to refine and release resources to the community, I recommend acceptance.

**Additional Comments On Reviewer Discussion:**

The authors actively addressed concerns and clarified details during the discussion, improving confidence in the reproducibility and relevance of their work. While questions remain about certain theoretical justifications and dataset details, the reviewers generally agree that the overall contribution stands.

---

### Decision · Program_Chairs · 2025-01-22

Accept (Poster)